# Circular synthesized CRISPR/Cas gRNAs for functional interrogations in the coding and noncoding genome

Martin Wegner[1†], Valentina Diehl[1†], Verena Bittl[1,2], Rahel de Bruyn[1], Svenja Wiechmann[1,3], Yves Matthess[1], Marie Hebel[1], Michael GB Hayes[4], Simone Schaubeck[1], Christopher Benner[4], Sven Heinz[4], Anja Bremm[1,2], Ivan Dikic[1,2,5,6], Andreas Ernst[1,3], Manuel Kaulich[1,5,6]*

[1]Institute of Biochemistry II, Goethe University Frankfurt - Medical Faculty, University Hospital, Frankfurt, Germany; [2]Buchmann Institute for Molecular Life Sciences, Goethe University, Frankfurt, Germany; [3]Project Group Translational Medicine & Pharmacology TMP, Fraunhofer Institute for Molecular Biology and Applied Ecology IME, Frankfurt, Germany; [4]Department of Medicine, University of California, San Diego, San Diego, United States; [5]Frankfurt Cancer Institute, Frankfurt am Main, Germany; [6]Cardio-Pulmonary Institute, Frankfurt am Main, Germany

*For correspondence:
kaulich@em.uni-frankfurt.de

[†]These authors contributed equally to this work

**Abstract** Current technologies used to generate CRISPR/Cas gene perturbation reagents are labor intense and require multiple ligation and cloning steps. Furthermore, increasing gRNA sequence diversity negatively affects gRNA distribution, leading to libraries of heterogeneous quality. Here, we present a rapid and cloning-free mutagenesis technology that can efficiently generate covalently-closed-circular-synthesized (3Cs) CRISPR/Cas gRNA reagents and that uncouples sequence diversity from sequence distribution. We demonstrate the fidelity and performance of 3Cs reagents by tailored targeting of all human deubiquitinating enzymes (DUBs) and identify their essentiality for cell fitness. To explore high-content screening, we aimed to generate the largest up-to-date gRNA library that can be used to interrogate the coding and noncoding human genome and simultaneously to identify genes, predicted promoter flanking regions, transcription factors and CTCF binding sites that are linked to doxorubicin resistance. Our 3Cs technology enables fast and robust generation of bias-free gene perturbation libraries with yet unmatched diversities and should be considered an alternative to established technologies.
DOI: https://doi.org/10.7554/eLife.42549.001

## Introduction

CRISPR/Cas has rapidly become the gold standard for unbiased high-throughput experiments, outperforming preexisting technologies such as RNAi (*Evers et al., 2016*; *Morgens et al., 2016*). A fundamentally important aspect of high-fidelity CRISPR/Cas screening is the quality of the gRNA library that is interrogated, with its diversity and distribution primarily influencing downstream experimental scales (*Sanson et al., 2018*). Conventionally used methods to generate gRNA libraries in pooled or arrayed formats include T4 ligase or homology-based cloning techniques, which require the PCR-based amplification of gRNA-encoding oligonucleotides as well as the presence of open plasmid DNA for successful gRNA sequence cloning (*Arakawa, 2016*; *Koike-Yusa et al., 2014*; *Ong et al., 2017*; *Schmidt et al., 2015*; *Shalem et al., 2014*; *Vidigal and Ventura, 2015*; *Wang et al., 2014*). Owing to these technical constraints, conventional libraries contain an unwanted PCR and cloning-dependent bias in their gRNA distribution that influences the experimental scale required for

statistically significant hit calling (*Shalem et al., 2014*; *Wang et al., 2014*). CRISPR libraries have become ubiquitously used in functional genomics efforts, underscoring relevance and utility of new PCR- and cloning-free technologies.

The rod-shaped filamentous phage M13 differs from other bacteriophages in that its genome-packaging capacity is variable and in that it is present as single-stranded (ss) DNA. Kunkel mutagenesis utilizes M13's  malleable coat and the ease of ssDNA purification from M13 phage and enables rapid site-specific mutagenesis to construct high-quality phage display libraries (*Handa and Varshney, 1998*; *Huang et al., 2012*; *Kunkel, 1985*; *Kunkel, 2001*). Kunkel mutagenesis has significantly contributed to the great success of phage display technologies (*Ernst et al., 2013*; *Sidhu, 2001*).

Here, we demonstrate the applicability of Kunkel mutagenesis in the generation of high-quality and high-fidelity CRISPR/Cas and RNAi gene perturbation reagents. In more detail, we developed a highly reproducible improved Kunkel mutagenesis technology that is designed to generate 3Cs CRISPR/Cas gRNA libraries robustly over a broad range of gRNA diversities. We demonstrate the high fidelity of 3Cs gRNA libraries by targeting all human DUBs and then determining their proliferative depletion phenotype, confirming previously known and discovering hitherto unknown DUB phenotypes. In an effort to enable unbiased screening within coding and noncoding regions, we encoded SpCas9 nucleotide preferences into a degenerated oligonucleotide and generated a highly complex CRISPR/Cas gRNA library (truly genome-wide (TGW)). Doxorubicin-positive selection screens with the TGW library in unperturbed human telomerase-immortalized retinal pigmented epithelial cells (hTERT-RPE1) were used to identify coding and noncoding regions, emphasizing the relevance of noncoding sequence elements in drug-resistance mechanisms. To enable high-content functional interrogations on a truly genome-wide scale, we introduce an optimized version of this library (oTGW). In summary, we establish the 3Cs technology as a robust alternative method for the generation of high-quality CRISPR/Cas gene perturbation libraries.

## Results

### Circular synthesized gRNAs are high-quality CRISPR/Cas reagents

In classical Kunkel mutagenesis (*Kunkel, 1985*; *Kunkel, 2001*), the circular ssDNA isolated from filamentous phage is hybridized with a complementary oligonucleotide that is extended and ligated to obtain a double-stranded DNA plasmid. As Kunkel mutagenesis is performed on ssDNA, we anticipated that it would be insensitive to the secondary DNA structures of viral sequence elements and therefore should enable the PCR and cloning-free generation of lentiviral gene perturbation reagents (*Huang et al., 2012*; *Kunkel, 1985*). We therefore hypothesized that the generation of lentiviral CRISPR/Cas gRNA libraries using circular ssDNA and Kunkel mutagenesis would reduce the coupling of gRNA diversity to gRNA distribution and would generate reagents of high quality (*Figure 1A*).

To demonstrate its general applicability to lentiviral CRISPR/Cas plasmids, we transformed *Escherichia coli* CJ236 bacteria with the commonly used pLentiGuide and pLentiCRISPRv2 plasmids (*Sanjana et al., 2014*), both of which contain a U6 promoter-controlled non-human targeting (NHT) placeholder gRNA followed by a SpCas9 tracrRNA sequence (*Doench et al., 2014*; *Sanjana et al., 2014*). Importantly, F-factor-containing CJ236 bacteria lack dUTPase ($dut^-$) and uracil-glycosylase ($ung^-$), and therefore tolerate the presence of deoxyuridine (dU) in genomic and plasmid DNA (*Kim and Wilson, 2012*). Superinfection of single-colony CJ236 culture with M13KO7 bacteriophage ($10^8$ cfu/mL) facilitated the generation of >30 µg of dU-containing circular ssDNA. Although circular ssDNA is identical in length to dsDNA, the circular ssDNA of lentiviral CRISPR/Cas plasmids migrated faster and appeared as a single band in gel electrophoresis (*Figure 1B*). Circular dU-ssDNA was hybridized with a gRNA-encoding complementary oligonucleotide that contained sequence homology regions (3Cs homology) at its 5′ and 3′ ends, and then extended and ligated with T7 polymerase and T4 ligase, respectively (*Figure 1A*). This resulted in heteroduplexed 3Cs DNA (3Cs-dsDNA), which were composed of dU-template ssDNA and deoxythymidine-containing newly synthesized complementary DNA that also includes the gRNA-encoding oligonucleotide (*Figure 1A*) (*Huang et al., 2012*; *Kunkel, 1985*; *Kunkel, 2001*). To gain insights into the oligonucleotide requirements and kinetics of 3Cs reactions, we tested different 3Cs homology

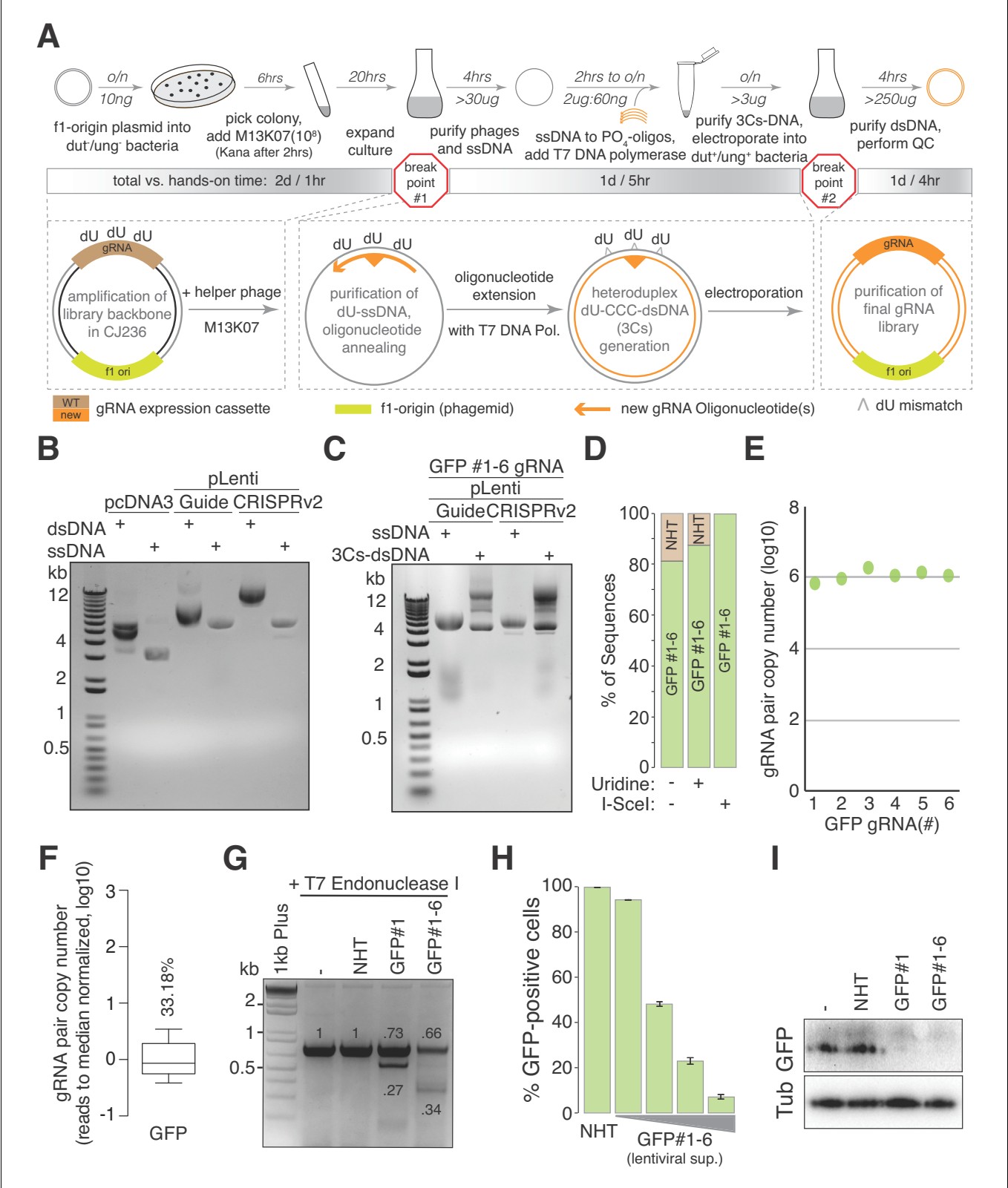

**Figure 1.** The 3Cs technology - covalently-closed-circular-synthesized (3Cs) CRISPR/Cas gRNA reagents. (**A**) The general 3Cs workflow. The individual steps of the protocol (grey arrows), time requirements (on top of arrow) and used or expected DNA yields (below arrow) are highlighted. Time requirements are separated by total versus hands-on time (grey scaled bars). Please note that the protocol contains two possible break points (red stop signs) at which purified phages can be stored at 4°C (break point #1) or bacterial pellets/purified plasmid DNA can be stored at −20°C (break point #2). *Figure 1 continued on next page*

*Figure 1 continued*

In more detail, f1-origin containing double-stranded CRISPR/Cas plasmids are converted to dU-containing circular ssDNA. Guide RNA sequence (orange triangle) containing oligonucleotides (orange arrows) are annealed to ssDNA, and extended and ligated by T7 DNA polymerase and T4 ligase, respectively. Heteroduplex dU-3Cs-DNA is transformed into base-excision-repair-sufficient bacteria to deplete template DNA (grey strand) and to amplify the newly synthesized DNA (orange) selectively. (B) Lentiviral CRISPR/Cas plasmids (pLentiGuide, pLentiCRISPRv2) and the mammalian cDNA expression plasmid pcDNA3 (positive control) were converted to dU-containing circular ssDNA and analyzed by gel electrophoresis. Although identical in size, circular ssDNA appears as a single band and migrates faster than the corresponding dsDNA form. (C) The lentiviral circular ssDNA of panel (B) was annealed with a pool of six oligonucleotides, encoding six GFP-targeting gRNAs, to generate a pool of 3Cs-dsDNA and analyzed by gel electrophoresis. A successful 3Cs in vitro reaction is indicated by three distinct 3Cs-dsDNA product bands (*Huang et al., 2012*). (D) Bar graph showing the degree of template remnants in the final 3Cs products in the presence and absence of additional Uridine in the phage culture medium as well as an I-SceI clean-up step. The gRNA libraries from panel (C) were sequenced by NGS before and after I-SceI restriction enzyme digest. Although the effect of Uridine is marginal, an enzymatic digest with I-SceI removes template plasmid remnants. (E, F) gRNA distribution displayed as raw read count data points (E) and normalized values in box plot format (F). The coefficient of variation was calculated by dividing the standard deviation by the mean of the library's read counts and is displayed as percentage above the box plot (F). Data were derived from the NGS data shown in panel (D). The final GFP-targeting 3Cs-gRNA library is free of sequence bias, as demonstrated by the low coefficient of variation of 33.18%, and by the uniform sequence distribution ((E), also see *Figure 1—figure supplement 1H*). (G) GFP-expressing hTERT–RPE1 cells were transduced with lentiviral 3Cs-gRNA constructs (non-targeting control gRNA (non-human target sequences (NHT)), a single GFP-targeting 3Cs-gRNA (GFP#1) or a pool of six GFP-targeting 3Cs-gRNAs (GFP#1–6)), and selected with puromycin before GFP gene editing was analyzed by T7 endonuclease I assay (*Guschin et al., 2010*). Individual band intensities were quantified (black numbers). An empty control (–) served as the reference. (H) A dose-dependent reduction of GFP fluorescence was determined by the flow cytometry of GFP-expressing hTERT–RPE1 cells and transduced with increasing volumes of lentiviral supernatant containing a pool of six GFP-targeting 3Cs-gRNAs (GFP#1–6). Error bars represent standard deviations (SDs) over three biological replicates ($n = 3$). (I) Immunoblot analysis of hTERT–REP1 cells treated as in panel (G) demonstrates that GFP-targeting 3Cs-gRNAs induce a 3- to 4-fold reduction in total GFP protein levels over three biological replicates ($n = 3$, for quantification see also *Figure 1—figure supplement 1I*).

DOI: https://doi.org/10.7554/eLife.42549.002

The following figure supplement is available for figure 1:

**Figure supplement 1.** Determining 3Cs parameters, I-SceI template remnant removal, and the GFP library.
DOI: https://doi.org/10.7554/eLife.42549.003

lengths of 10, 13, 15, and 18 nucleotides, performed a 3Cs reaction time series, and demonstrated that 18 nucleotides of homology (above 45°C annealing temperature) and 2 hr of 3Cs reaction time were sufficient (*Figure 1—figure supplement 1A–C*) (*Kunkel, 2001*).

Using rule set 2 (RS2) (*Doench et al., 2014*, *Doench et al., 2016*), we designed six GFP-targeting gRNA sequences and extended them by 5′ and 3′ 3Cs homology. Synthesized gRNA-encoding oligonucleotides were hand-pooled in equimolar ratios, phosphorylated and used in a 1:5 ratio (2 μg ssDNA to 60 ng oligonucleotide) to generate heteroduplex dU-3Cs-sDNA (*Figure 1C*). To remove NHT/dU-containing template and to amplify the gRNA-encoding complementary strand, 3Cs products were column-purified and transformed in $dut^+/ung^+$ bacteria. Bacterial clones were grown and their plasmid DNA Sanger sequenced, revealing that 81% of pLentiGuide and 82% of plenti-CRISPRv2 contained GFP-targeting gRNAs (*Figure 1D* and *Figure 1—figure supplement 1D*). To test whether dU supplementation reduces the amount of NHT-containing template plasmid by improving dU-incorporation during ssDNA production, CJ236 culture medium was supplemented with 2.5 μM dU. In addition, the gRNA placeholder sequence of pLentiGuide and plentiCRISPRv2 was changed to contain an I-SceI restriction enzyme recognition site. Although the effect of increased dU concentrations was negligible, I-SceI-mediated removal of wildtype plasmids reduced their level to below our detection limit (*Figure 1D* and *Figure 1—figure supplement 1D–F*). We performed next-generation sequencing (NGS) on the plentiCRISPRv2 sample with an average read count of 1.15 million per GFP sequence and identified a wildtype rate of below 0.3% in the absence of any apparent sequence bias (*Figure 1E* and *Supplementary file 1*). A one-sided Chi-squared test for goodness of fit identified a uniform distribution (p=0.1) of all six gRNA sequences. The uniform gRNA distribution was supported by a low coefficient of variation (CV) of 33.18% and an area under the curve (AUC, Lorenz curve) of only 0.56 (*Figure 1E–F* and *Figure 1—figure supplement 1G*).

To test for the cellular functionality of 3Cs gRNAs, we used the plentiCRISPRv2 GFP-targeting 3Cs gRNA constructs to generate infectious lentiviral particles and transduced GFP-positive human telomerase-immortalized retinal pigmented epithelial (hTERT–RPE1) cells. Seven days post-transduction, we performed a T7 Endonuclease I assay and observed robust GFP gene editing, both by a single GFP-targeting 3Cs gRNA (3Cs-gRNA) and by the pool of six 3Cs-gRNAs, whereas un-transduced

(–) and an NHT control gRNA failed to edit the GFP gene (*Figure 1G*). GFP gene editing translated to a lentiviral dose-dependent loss of GFP protein when analyzed by fluorescence flow-cytometry and immunoblotting (*Figure 1H–I* and *Figure 1—figure supplement 1H*). Taken together, we demonstrate that the 3Cs technology enables the rapid and cloning-free generation of high-quality single and pooled CRISPR/Cas gRNAs.

## 3Cs uncouples sequence diversity from sequence distribution

The absence of PCR amplification and cloning steps, in combination with the uniform distribution of the six GFP-targeting 3Cs-gRNAs, led us to reason that 3Cs may uncouple sequence diversity from sequence distribution during gRNA library generation. To test this hypothesis, we designed six degenerated 3Cs oligonucleotides with increasing numbers of randomized nucleotides to mimic gRNA sequence pools with diversities ranging from 256 to 262,144 individual sequences (*Figure 2A*). The six pools were applied in parallel 3Cs syntheses on a dU-ssDNA template of pLenti-CRISPRv2 (*Figure 2B* and *Figure 2—figure supplement 1A*). Independent of an oligonucleotide's diversity, NGS and computational analyses identified all of the individual sequences and uniform distributions with area under the curve values between 0.6 and 0.73 (*Figure 2—figure supplement 1B* and *Supplementary file 2*). Despite the uniform distribution, we observed a prominent cytosine (C) bias in the randomized libraries, with C contents of above 40% within the top 10% of the most abundant gRNAs (*Figure 2C*). We reasoned that the C bias is probably due to incomplete phosphoramidite mixing during oligonucleotide synthesis and should therefore be absent from gRNA libraries containing nonrandom gRNA sequences (*Ellington and Pollard, 2009*). To test this hypothesis, we designed and generated a nonrandom 3Cs-gRNA library targeting all 105 human DUBs, each with three gRNAs (DUB library). NGS and nucleotide content analysis confirmed our hypothesis and revealed the absence of C bias from the nonrandom DUB library (*Figure 2C* and *Supplementary file 3*). To correct the randomized libraries for the C bias, we determined the individual nucleotide frequency at every randomized position and used these frequencies to normalize the original read counts, leading to improved AUC values and sequence distributions (*Figure 2D* and *Supplementary file 2*) and further confirming the uncoupling of sequence diversity and distribution in 3Cs reactions. Taken together, these findings lead us to conclude that 3Cs is a robust technology that uncouples sequence distribution from sequence diversity and, therefore, is a powerful alternative technology to conventional gRNA cloning methods for generating gRNA libraries.

## 3Cs-gRNA libraries are of high fidelity: the proliferative essentiality of human DUBs

Next, we investigated the performance of 3Cs-gRNA reagents in cellular screenings. To do so, we generated infectious lentiviral particles of the 3Cs-gRNA DUB library and applied them to a proliferation screen in non-transformed hTERT–RPE1 cells in biological duplicates (multiplicity of infection (MOI) 0.2, coverage 1,000). Two days after lentiviral transduction, cells were either collected (day 0, reference time point) or selected by puromycin and kept in culture for 11 days (day 11) or 21 days (day 21) in cycling conditions representing at least a 1,000-fold library coverage (*Figure 3A*). Cells collected at day 0, 11, or 21 were subjected to genomic DNA extraction and amplicon-based NGS library preparation, as has been reported previously (*Doench et al., 2016*; *Koike-Yusa et al., 2014*). We performed single-read sequencing on an Illumina NextSeq500 with an averaged read count per gRNA of above 35,000 across all biological samples and replicates (*Supplementary file 4*). As in previously reported CRISPR analysis algorithms, and to enable a comparison of individual time points, we summed all individual gRNA read counts per gene and normalized each gene read count per sample to the total number of read counts within that sample (*Supplementary file 4*) (*Li et al., 2014*; *Spahn et al., 2017*). In line with reports of the high experimental reproducibility of CRISPR/Cas screenings (*Evers et al., 2016*; *Morgens et al., 2016*), we determined $R^2$ values of 0.95, 0.88, and 0.90 for time points day 0, 11, and 21, respectively (*Figure 3—figure supplement 3A–C*). Spearman correlation and Shapiro–Wilk confidence tests revealed correlations of above 0.88 and p-values of below 0.001, respectively, demonstrating a high experimental confidence in the level of gRNA representation (*Figure 3B* and *Figure 3—figure supplement 3A–C*) (*Shapiro and Wilk, 1965*). To analyze the reproducibility of our screen in terms of the level of gene phenotypes, we applied MAGeCK and PinAPL-Py, two established algorithms for the

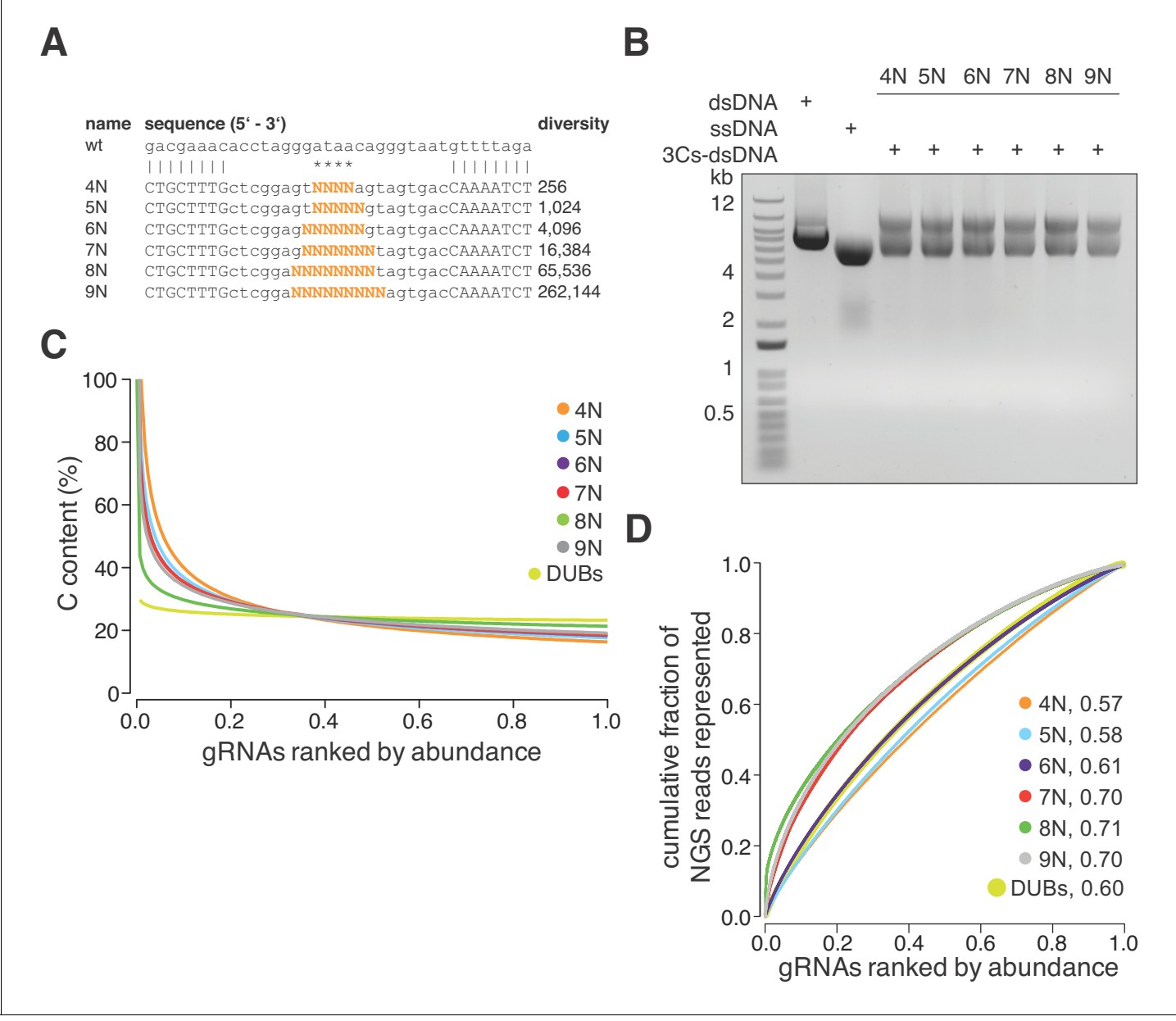

**Figure 2.** 3Cs is a robust technology that uncouples sequence diversity from sequence distribution. (**A**) To determine the sequence distribution of 3Cs-gRNA libraries with increasing gRNA diversity, an increasing number of randomized nucleotides (orange) were incorporated into 3Cs oligonucleotides to mimic gRNA diversities ranging from 256 to 262,144 sequences (4–9N libraries). A range of four to nine randomized nucleotides (orange) were introduced into an NHT gRNA sequence. Randomization of the central nucleotides ensures the replacement of the template I-SceI restriction site in order to prevent the digestion of correctly synthesized 3Cs synthesis products. (**B**) The 3Cs synthesis products of the combination of randomized primers and pLentiGuide were resolved by gel electrophoresis. (**C**) The Scatter plot displays ranked gRNA abundances per library against the gRNA cytosine content (C). The gRNA libraries that are shown are derived from (**A**) and (**B**) and the library with gRNAs targeted against DUBs (DUBs library). All libraries were processed by I-SceI-dependent removal of template plasmid remnants and subjected to NGS and computational analysis. Importantly, all gRNA libraries were complete, irrespective of their individual gRNA diversity. However, the partially randomized gRNA libraries displayed a strong C bias within the most abundant gRNA sequences. In fact, the top 5% of most abundant gRNAs had a C content of above 60%. The DUBs library did not show this C bias, strongly suggesting incomplete phosphoramidite mixing during oligonucleotide synthesis as the main cause of the C bias. (**D**) Lorenz curves displaying the cumulative fraction of represented NGS reads versus the gRNAs ranked by abundance of each partially randomized (4–9N) and nonrandomized (DUB) library revealed a uniform distribution of gRNA sequences. Area under the curve values (AUC, number next to library name) confirm the uniform gRNA distribution of these libraries and demonstrate that 3Cs uncouples sequence diversity from sequence distribution.

DOI: https://doi.org/10.7554/eLife.42549.004

*Figure 2 continued on next page*

*Figure 2 continued*

The following figure supplement is available for figure 2:

**Figure supplement 1.** Quality control and gRNA distributions of the randomized libraries.
DOI: https://doi.org/10.7554/eLife.42549.005

analysis of CRISPR/Cas screens, to raw gRNA read counts of both replicates and calculated aggregated positive and negative proliferation phenotypes by means of $\log_2$-fold changes with associated p-values (*Figure 3C–E* and *Figure 3—figure supplement 3D–H* and *Supplementary file 5–6*) (*Li et al., 2014*; *Spahn et al., 2017*). Consistent over both time points, cells depleted of USP28 or BRCC3 proliferated more rapidly than cells harboring non-human target sequences (NHTs), identifying both as negative regulators of hTERT–RPE1 proliferation (*Figure 3C–D* and *Figure 3F* and *Supplementary file 4–5*). By contrast, cells that were depleted of PSMD14, USP7 or COPS6 proliferated less rapidly than cells harboring non-human target sequences (NHTs), identifying them as positive regulators of hTERT–RPE1 proliferation (*Figure 3C–D* and *Figure 3F* and *Supplementary file 4–5*).

CRISPR/Cas drop out screens are performed with varying experimental durations, ranging from 5 to 15 days (*Joung et al., 2017*; *Potting et al., 2018*). However, recent work demonstrates that CRISPR/Cas induces a $G_1$ phase arrest in p53 proficient hTERT–RPE1 cells that impacts hit calling (*Haapaniemi et al., 2018*), suggesting that later screening time points are beneficial for hit calling. Indeed, when comparing normalized gene ranks, we observed a trend of increased phenotype resolution among negative gene ranks over time, although this effect was largely absent from positive gene ranks (*Figure 3—figure supplement 1I* and *Figure 3—figure supplement 2* and *3*). This effect has been reported previously and can potentially be explained by the disproportional assay window of positive and negative cell proliferation phenotypes, leading to a higher phenotypic resolution among negative proliferative effects (*Shalem et al., 2014*; *Wang et al., 2014*).

Stable and robust proliferative phenotypes are time-independent, so the phenotype resolution enhances over time (*Figure 3—figure supplement 3I*). However, multiple mechanisms and cellular backgrounds can influence phenotype strength, timing and orientation. To identify time-dependent phenotypes, we analyzed the MAGeCK-derived $\log_2$ fold changes with associated p-values for time points day 11 and day 21, and identified genes whose deletion phenotype significantly changed between day 11 and day 21 (*Figure 3E–F*). Depletion of USP28 and USP46 induced the strongest positive change, whereas deletion of USP22, USP48 or TNFAIP3 induced the most significant negative change, in phenotype between day 11 and day 21, suggesting a time-dependent absence of compensatory mechanisms to accommodate an early loss-of-function phenotype (*Figure 3E–F* and *Figure 3—figure supplement 3I*). In order to validate our findings, we chose two positive and negative proliferation-inducing DUBs, generated lentiviral supernatant to deliver shRNA sequences targeting the selected DUBs, and transduced hTERT–RPE1 cells. Over the course of the 2 weeks after transduction, we measured cell numbers by AlamarBlue staining. When compared to negative- (Luciferase) and positive-control (Plk1) shRNA sequences, depletion of USP28 and BRCC3 induced a rapid positive proliferation effect (*Figure 3G*). By contrast, depletion of USP7 and COPS6 induced an instant and strong negative proliferation effect (*Figure 3G*), validating the gRNA-mediated knockout phenotypes. Collectively, our analysis demonstrates the quality and fidelity of 3Cs reagents in functional genomics applications.

## 3Cs is versatile and generates arrayed and pooled 3Cs-shRNA reagents

Owing to its versatility, CRISPR/Cas technology has become the method of choice for gene perturbation experiments, yet classical short hairpin RNAs (shRNA, RNAi) are still widely used. However, shRNA oligonucleotides contain complementary sequences that form stable secondary structures that render the generation of shRNA reagents inefficient (*McIntyre and Fanning, 2006*). A crucial step in our improved Kunkel mutagenesis technology is the denaturation of the gRNA-encoding oligonucleotides and their subsequent annealing to template ssDNA (see 'Materials and methods' section). Owing to this denaturation and annealing step, we anticipated that improved Kunkel mutagenesis would circumvent the secondary structures of shRNA oligonucleotide and enable the generation of shRNA reagents. We chose pLKO.1 (*Stewart et al., 2003*), one of the most widely

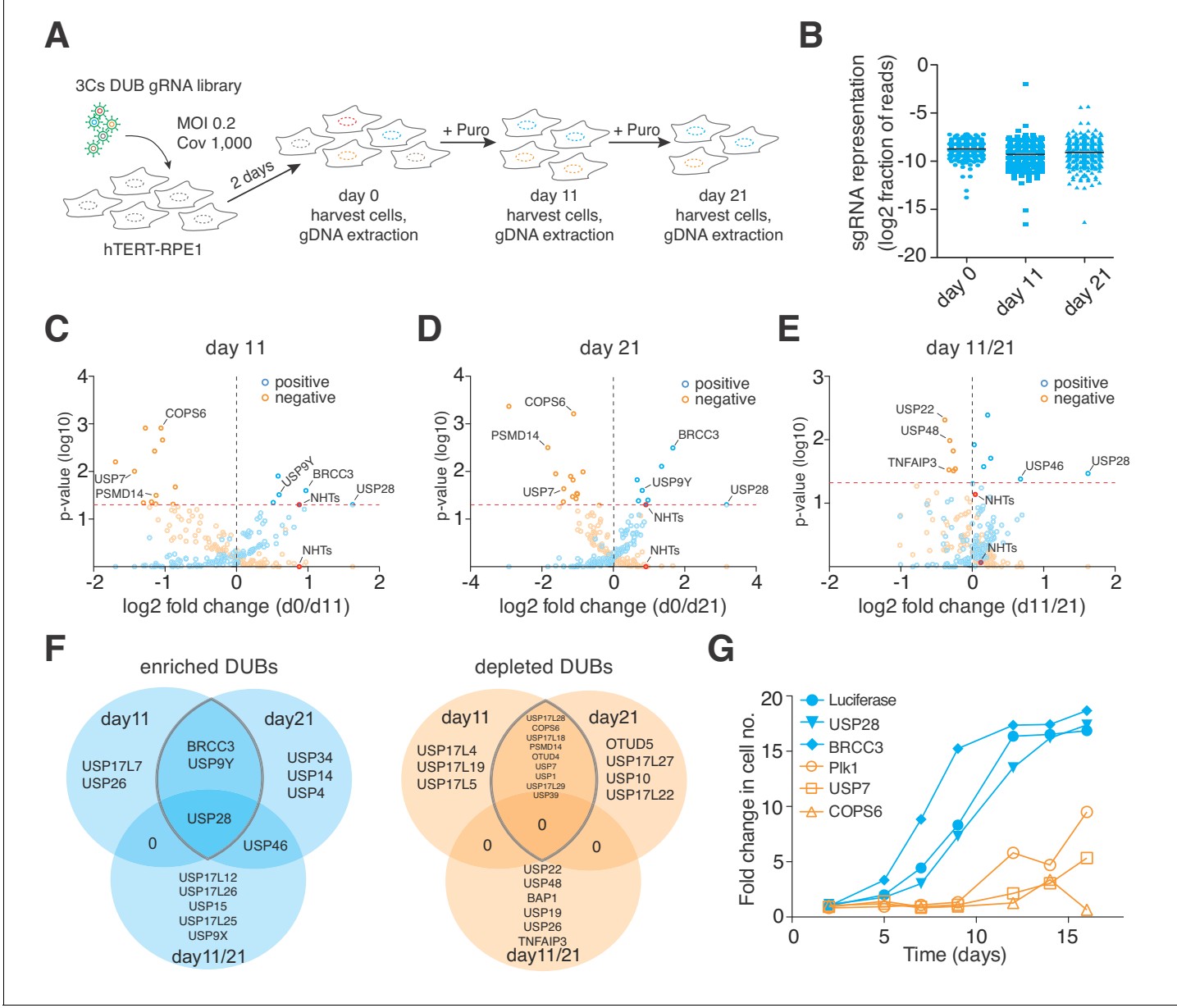

**Figure 3.** 3Cs reagents are of high fidelity — the essentiality of human DUBs for cell fitness. (**A**) Schematic of the performed CRISPR screen. Highlighted are the experimental conditions under which the screen was performed (MOI of 0.2, library coverage of 1,000). In brief, hTERT–RPE1 cells were transduced with lentivirus for 48 hr in duplicates, after which the cells of one duplicate were harvested (day 0) to extrapolate the baseline gRNA distribution. Simultaneously, cells of the second duplicate were subject to puromycin selection for 11 days, after which time all cells were harvested (day 11), counted and plated back in low density to the original library representation of 1,000-fold coverage. Plated cells remained in cycling conditions until day 21, when all cells were collected (day 21). After harvesting the cells, their genomic DNA was extracted and processed for gRNA NGS. (**B**) Graph showing the distribution of individual sgRNAs. Means ± standard deviation are highlighted. (**C–E**) Volcano plots visualizing $\log_2$ fold changes of gene phenotypes and their associated p-values. Data are derived from MAGeCK analyses, corresponding to day 11 (**C**), day 21 (**D**) and day11/21 (**E**). The dashed red line shows p=0.05 with points above the line having p<0.05 and points below the line having p>0.05. Data points with p>0.05 are displayed as translucent symbols. Genes of interest are highlighted. (**F**) Venn diagram of significantly enriched (blue) or depleted (orange) DUBs. The time point overlap visualizes DUB genes with time independent (overlap of three) and time dependent (overlap of two) proliferation phenotypes. (**G**) Fold increase in cell number after shRNA-mediated depletion of target genes. Data are means of duplicates.

DOI: https://doi.org/10.7554/eLife.42549.006

The following figure supplements are available for figure 3:

**Figure supplement 1** The 3Cs DUB gRNA screens are highly reproducible.

DOI: https://doi.org/10.7554/eLife.42549.007

*Figure 3 continued on next page*

*Figure 3 continued*

**Figure supplement 2.** 3Cs facilitates the generation of shRNA libraries.
DOI: https://doi.org/10.7554/eLife.42549.008

**Figure supplement 3.** A 3Cs E2 shRNA library.
DOI: https://doi.org/10.7554/eLife.42549.009

used lentiviral and shRNA-expressing plasmids to generate circular dU-ssDNA. Similar to circular ssDNA of CRISPR/Cas plasmids, the circular ssDNA of pLKO.1 migrated as a single band in agarose gel electrophoresis (*Figure 3—figure supplement 2A*). Next, we designed a GFP-targeting 3Cs shRNA (3Cs-shRNA) primer consisting of 5′ and 3′ 3Cs homology and two complementary GFP–shRNA sequences separated by a six-nucleotide hairpin sequence (*Figure 3—figure supplement 2B*). We performed two parallel 3Cs reactions using 60 ng and 120 ng of ssDNA, and both reactions yielded the characteristic 3Cs-DNA band pattern with no major difference in bacterial transformation efficiency between the two tested scales (*Figure 3—figure supplement 2C*).

To demonstrate the successful integration of the GFP–shRNA sequence into pLKO.1, we amplified single bacterial clones carrying 3Cs-DNA of shRNA reactions and analyzed their plasmid DNA by SANGER sequencing. This resulted in the expected GFP–shRNA sequence and the absence of adjacent nucleotide changes (*Figure 3—figure supplement 2D*), from which we concluded that 3Cs is a versatile technology that generates high-quality gRNA and shRNA reagents. To demonstrate 3Cs-shRNA fidelity, we generated infectious lentiviral particles of the GFP-targeting 3Cs-shRNA and transduced GFP-positive hTERT–RPE1 cells. Strikingly, 96 hr after lentiviral transduction, we observed a reduction in GFP-fluorescence, confirming the functionality of the 3Cs-shRNAs in cells (*Figure 3—figure supplement 2E*). Moreover, we investigated the performance of improved Kunkel mutagenesis in generating 3Cs-shRNA libraries. On the basis of the principles described above, we designed a 3Cs-shRNA library targeting all human ubiquitin-conjugating E2 enzymes (E2s), each with two shRNAs (*Supplementary file 7*). To generate the library, individually synthesized oligonucleotides were pooled in equimolar ratios and applied to a pooled 3Cs reaction. The resulting products were resolved by gel electrophoresis (*Figure 3—figure supplement 3A*). Like the I-SceI-mediated depletion of wildtype remnants from CRISPR/Cas 3Cs-gRNA constructs, a Bsu36I restriction enzyme clean-up step removed pLKO.1 wildtype remnants, and SANGER sequencing of the final E2 3Cs-shRNA library (E2.2) confirmed a randomization of forward- and reverse-complement shRNA sequences (*Figure 3—figure supplement 3B–C*). To determine the E2 3Cs-shRNA distribution more accurately, we performed NGS sequencing with an average shRNA read count of >8,300 and determined a wildtype remnant level of 0.04%, a CV of 37.9% and an AUC of 0.68, demonstrating an almost uniform distribution (*Figure 3—figure supplement 3D–E*). To correlate 3Cs-shRNA abundance and the distribution of the E2 3Cs-shRNA libraries before and after Bsu36I enzyme digest, we determined the ratios of their respective normalized read counts. Importantly, all ratios were close to one and lined up close to the respective diagonal with a linear regression $R^2$ of 0.9687 (*Figure 3—figure supplement 5F*), demonstrating a high correlation of individual data points and no influence of the Bsu36I digest on 3Cs-shRNA sequence distribution. In summary, this demonstrates that our 3Cs technology can be adapted to generate high-quality shRNA reagents in single and pooled formats.

## A partially randomized 3Cs gRNA library to target the coding and noncoding genome simultaneously

The 3Cs method does not require the PCR-amplification of gRNA-encoding oligonucleotides, is free of conventional cloning steps and uncouples sequence diversity from sequence distribution. Thus, we hypothesized that 3Cs gRNA library diversity is mostly limited by the number of distinguishable oligonucleotides within a 3Cs reaction and the subsequent bacterial electroporation efficiencies. Limitations in electroporation efficiencies can be overcome by accumulating the individual efficiencies of multiple parallel reactions, as routinely performed to amplify phage libraries with diversities beyond $10^9$ (*Smith and Scott, 1993*). The number of distinguishable oligonucleotides is limited by the capacity of synthetic oligonucleotide synthesis, rendering truly genome-wide gene perturbation libraries unfeasible.

Previously identified SpCas9 nucleotide preferences included a preference for 3′ pyridine bases whereas thymidine nucleotides are disfavored (*Doench et al., 2014*; *Doench et al., 2016*). In an exploratory effort, we translated SpCas9 gRNA nucleotide preferences into a degenerated oligonucleotide sequence (truly genome-wide, TGW) of 20 nucleotides, representing a theoretical diversity of $7.3 \times 10^{10}$ (*Figure 4A*) and maximally targeting $1.65 \times 10^{7}$ sites in the human coding and noncoding genome (*Figure 4A* and *Supplementary file 8*). As mentioned above, randomized positions in DNA oligonucleotides can contain a strong single nucleotide cytosine bias, we therefore used hand-mixed phosphoramidite pools to generate this oligonucleotide pool. In eight parallel large-scale 3Cs reactions (each involving 20 µg ssDNA and 600 ng oligonucleotide), we applied this oligonucleotide to dU-ssDNA of pLentiGuide and pLentiCRISPRv2 and resolved the 3Cs products by gel electrophoresis (*Figure 4B*). Importantly, we note that the amplification of this degenerated library is limited by the number of transformed bacteria. As complete generation of this reagent is currently unfeasible, we limited our efforts to eight parallel electroporation reactions and achieved a cumulative transformation efficiency of $1.2 \times 10^{10}$, accounting for ~16% of TGW sequences, assuming a stringent uniform sequence distribution. In order to approximate the gRNA distribution, we generated 14.4 million NGS reads and found 94.19% to be unique (*Figure 4C* and *Supplementary file 9*). We went on to extract the nucleotide frequencies for each gRNA position from TGW NGS reads, translated them to IUPAC nomenclature, and identified the identical degenerated sequence that we initially applied in the form of the degenerated oligonucleotide pool (*Figure 4D*). Furthermore, the distribution of TGW read counts had a CV of 0.26% and an AUC of 0.52, suggesting a nearly uniform distribution of sequenced and represented gRNA sequences (*Figure 4E–F*) (*Makowski and Soares, 2003*).

The 3Cs technology enables the generation of gRNA libraries with sequence diversities exceeding those that can be captured by coverage-based screenings. Being aware of the coverage limitations, we explored a TGW library screen in the context of a strong positive selection pressure. In hTERT–RPE1 cells, doxorubicin induces a robust, irreversible and dose-dependent reduction of cell viability within 4 days (*Figure 5—figure supplement 1A*). We therefore generated $5.5 \times 10^{8}$ infectious lentiviral particles of the TGW library and applied them to screen for resistance to doxorubicin (*Figure 5A–B*). In three biologically independent experiments, we transduced a total of $5.5 \times 10^{8}$ hTERT–RPE1 cells with a MOI of 1, added 1 µM doxorubicin 7 d post transduction and replaced the media every 7 d for 21 consecutive days. Cells that survived the treatment were harvested and their genomic DNA was extracted for NGS (*Figure 5B*). Although the experimental reproducibility was low (0.004%), we identified an experimental overlap of 4,232 gRNAs, with associated Spearman ranking and Pearson correlations of above 0.75 (*Figure 5C* and *Figure 5—figure supplement 1B*). To validate these sequences, we designed and generated a new 3Cs-gRNA validation library consisting of the identified 4,232 gRNAs and repeated the doxorubicin resistance screen with established experimental parameters (coverage of 1,000 and MOI of 0.2) (*Figure 5—figure supplement 2A–B*). As a result, we reidentified 2,716 gRNAs of which 795 were more than two-fold enriched after 21 d of doxorubicin treatment when compared to the untreated control (*Figure 5D* and *Supplementary file 11*). In order to map the 795 gRNA sequences to a location within the human genome, we applied Cas-OFFinder and used the Ensembl, ENCODE, Roadmap Epigenomics and Blueprint databases for sequence annotation (*Bernstein et al., 2010*; *Dunham et al., 2012*; *Fernández et al., 2016*; *Zerbino et al., 2018*). We identified seven gRNAs to target five genes (PDE8B, AVPR2, CYSLTR2, IL3RA, and POLE2), of which PDE8B and AVPR2 were targeted by two gRNAs, and a single gRNA sequence matched a noncoding location within chromosome 8 (chr8:93022800) (*Figure 5E* and *Figure 5G* and *Supplementary file 12–13*). The coding hits that we identified included CysLTR2, a Leukotriene C4 G-protein-coupled eicosanoid receptor that was recently reported to induce doxorubicin resistance by abolishing the accumulation of reactive oxygen species (*Dvash et al., 2015*). To validate CysLTR2 as a doxorubicin-resistance inducing hit, we chemically inhibited CysLTR2 with increasing concentrations of Bay-CysLT2 or Bay-u9773 in the presence of doxorubicin and quantified cell viability by AlamarBlue staining. Importantly, both drugs reverted the doxorubicin-induced toxicity in a dose-dependent manner (*Figure 5F*), suggesting that the loss of CysLTR2 causes doxorubicin resistance.

To account for sequence differences between rRNAs from hTERT–RPE1 cells and the reference genome (GRCh38.86), as well as SpCas9 off-target activity (*Cho et al., 2014*; *Hsu et al., 2013*; *Pattanayak et al., 2013*), we extended our computational analysis by allowing up to two

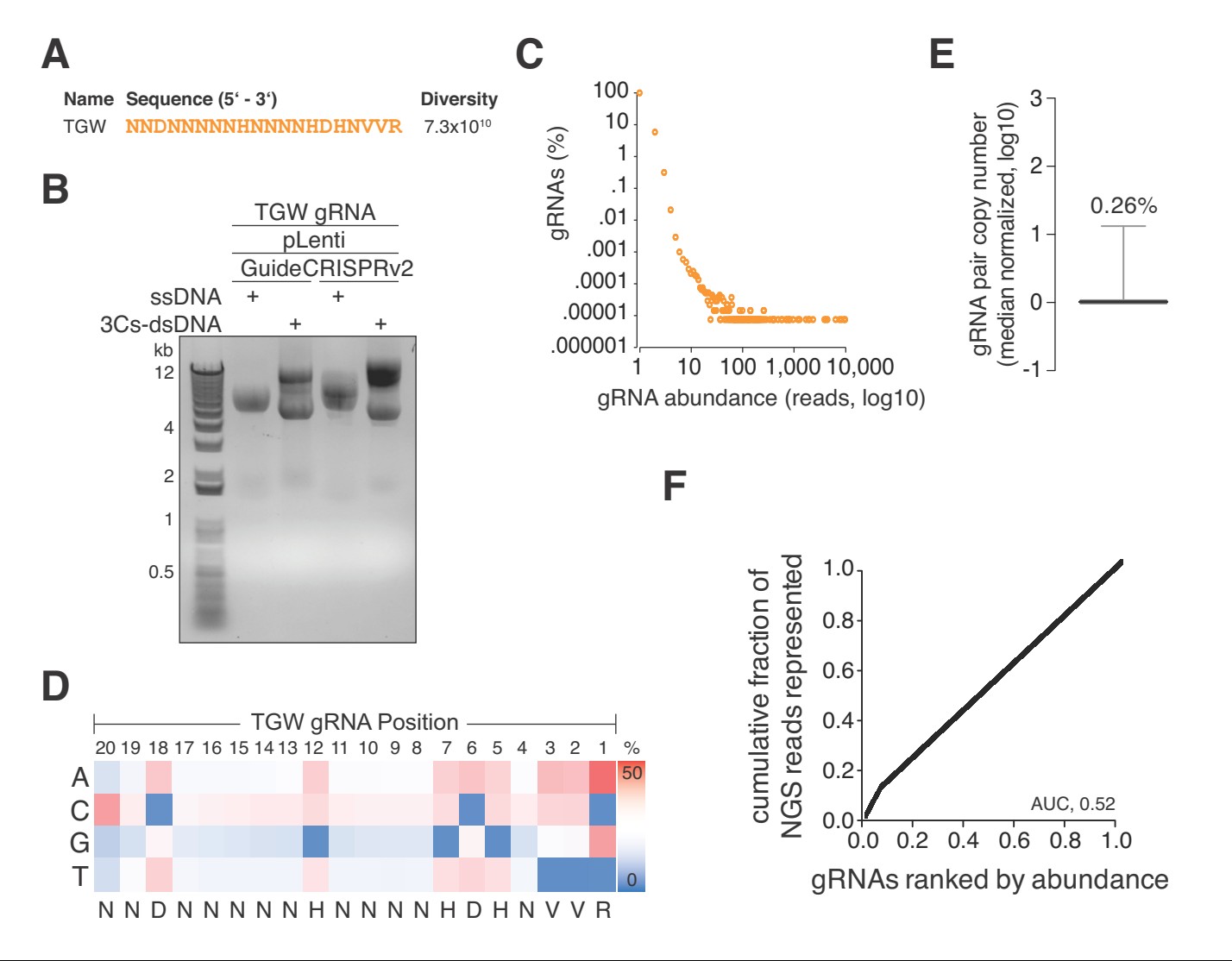

**Figure 4.** A truly genome-wide (TGW) CRISPR/Cas 3Cs-gRNA library to interrogate the coding and noncoding genome. (**A**) Previously reported SpCas9 nucleotide preferences were translated into a degenerated oligonucleotide sequence (TGW) representing a total sequence diversity of $7.3 \times 10^{10}$ (**Doench et al., 2014**). (**B**) The TGW oligonucleotide shown in panel (**A**) was used in a 3Cs reaction on template ssDNA derived from pLentiGuide and plentiCRISPRv2 plasmids to generate 3Cs-dsDNA, which was analyzed by gel electrophoresis. (**C**) Scatter plot visualizing TGW library NGS data from 14,448,469 total reads. Displayed are the $\log_{10}$ values of gRNA abundance (reads) against the $\log_{10}$ of the respective percentage of identified TGW gRNAs. 94.2% of all identified gRNAs were found only once (see also **Supplementary file 9**). (**D**) High-throughput sequencing data from panel (**C**) were used to compute the nucleotide frequency at each gRNA nucleotide position in order to determine the nucleotide profile of the TGW library. The identified nucleotide frequencies closely resemble the pattern of the degenerated TGW oligonucleotide from panel (**A**). Color code represents nucleotide frequency as indicated by the color gradient on the right. (**E**) Box plot of TGW gRNA distribution with data derived from panel (**C**). The coefficient of variation of 0.26% suggests a uniform distribution of represented sequences. (**F**) The gRNA distribution of the TGW library as derived from panel (**C**) plotted as a Lorenz curve. TGW NGS data derived from pane; (**C**). The area under the curve (AUC) of 0.52 suggests a uniform distribution of gRNA sequences.

DOI: https://doi.org/10.7554/eLife.42549.010

mismatches during Cas-OFFinder-based target sequence identification. As expected, the number of gRNAs that could be mapped to the reference genome increased to 192 and 222 for coding and noncoding target sites, respectively, accounting for 50.3% of the 795 gRNAs (**Figure 5E** and **Supplementary file 12**). Interestingly, when mismatches are allowed, we identified three gRNAs that targeted two different coding positions within the AKAP6 gene (chr14:32671632, chr14:32784395), as well as five different gRNAs that targeted the exact same coding position within

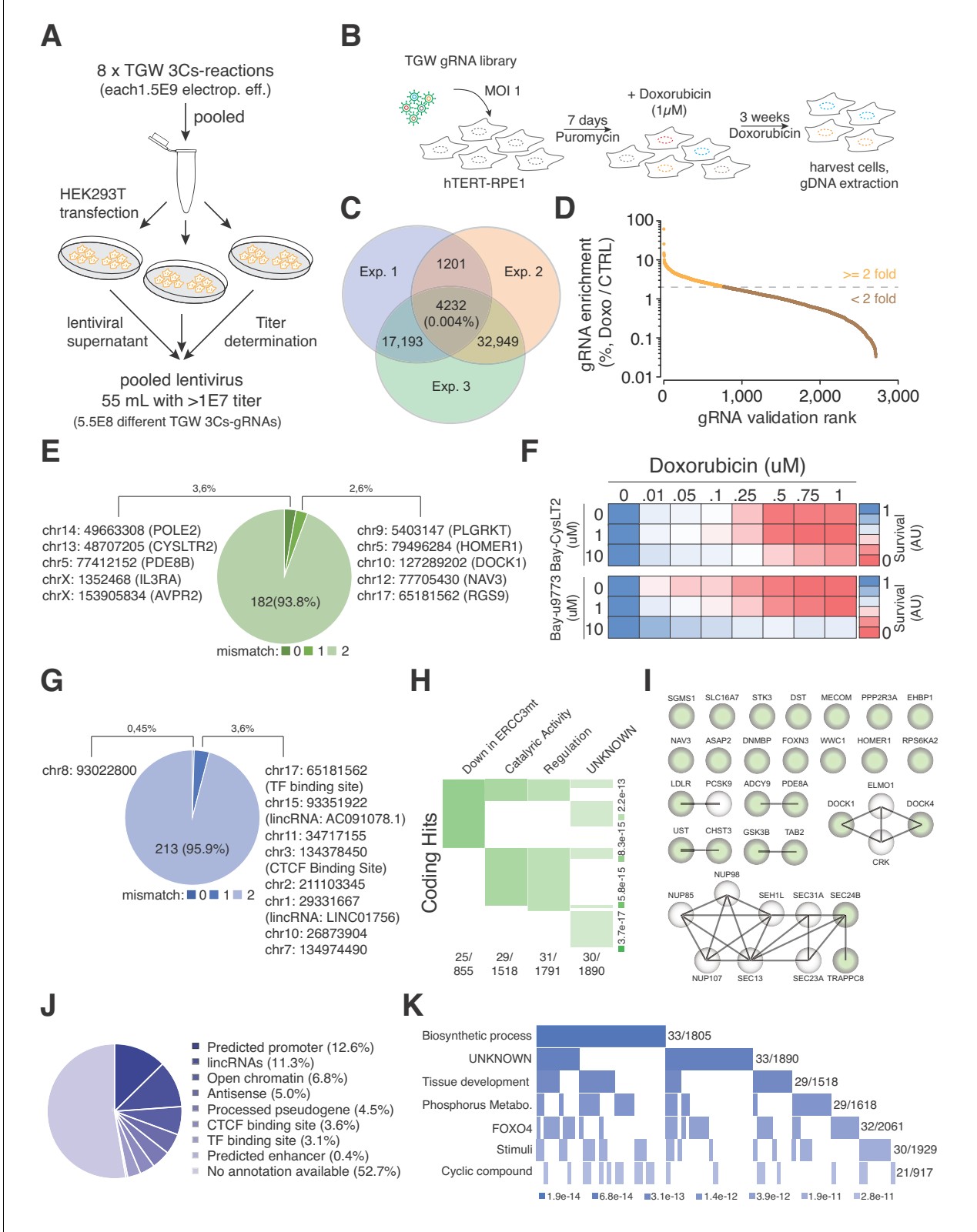

**Figure 5.** TGW-based identification of coding and noncoding sequences that are associated with doxorubicin resistance. (**A**) Scheme illustrating the workflow used to generate the pooled lentivirus of the TGW library. The DNA of eight independent TGW 3Cs syntheses was pooled and used to transfect HEK293T cells to produce $5.5 \times 10^8$ infectious lentiviral particles. (**B**) Experimental workflow of the doxorubicin screen in hTERT–RPE1 cells. hTERT–RPE1 cells were transduced with TGW lentivirus with an MOI of 1, selected with puromycin for 7 days, and treated with 1 μM doxorubicin. After

*Figure 5 continued on next page*

*Figure 5 continued*

three weeks of continuous doxorubicin treatment, all surviving cells were collected and processed for further analysis. (C) Genomic DNA derived from three independent experiments (*n* = 3), performed according to the scheme illustrated in panel (B), was used to perform NGS and gRNA sequence identification. Computational analysis identified an experimental overlap of 4,232 gRNAs (see also *Supplementary file 10*). (D) A 3Cs library containing the experimental overlap of 4,232 gRNAs (the validation library) was generated and screened with an experimental coverage of 1,000 and an MOI of 0.1 (similar to the workflow shown in panel (B); see also *Figure 5—figure supplement 2*). NGS of all surviving cells and computational analysis identified 795 gRNAs that were enriched more than two-fold (orange) when compared to an untreated control. (E) Pie chart visualizing the distribution of coding target regions with respect to SpCas9 off-target rate (0 to 2 mismatches). A total of 192 gRNAs (22.38% of 795 gRNAs) could be mapped to coding regions. Color code represents degree of nucleotide mismatch. (F) Chemical inhibition of cells rescued by CysLTR2 from doxorubicin-mediated toxicity. hTERT–RPE1 cells were treated for 4 d with increasing concentrations of doxorubicin and two chemical inhibitors of CysLTR2 (Bay-CysLT2 and Bay-u9773) before cellular viability was determined by AlamarBlue assays. Averaged values over three biological replicates (*n* = 3) in arbitrary units (AU) are displayed. (G) Pie chart visualizing the distribution of noncoding target regions with respect to SpCas9 off-target rate (0 to 2 mismatches). A total of 222 gRNAs (27.92% of 795 gRNAs) could be mapped to noncoding regions. Color codingshows the degree of nucleotide mismatch. (H) Molecular signature analysis of coding gRNA target sites identifies a set of genes that are downregulated in cells expressing mutant forms of ERCC3 as the top hit. From among the 178-coding gRNA target site-associated genes, 25 genes are part of the ERCC3 group (which has a total of 855 genes) with high confidence (p=3.7e-17). (I) A list of the 25 'down in ERCC3 mutated cells' genes (light green), as well as their known first- and second-degree interacting genes (grey), identifies cytokinesis (DOCK1/4 genes) and vesicle transport (SEC24B/TRAPPC8 genes) gene interactions. Interaction data adapted from String 10.5. (J) Pie chart visualizing the distribution of noncoding gRNA target site annotations, including their frequency (as percentages of total noncoding hits). Please note: for 52.7% of all noncoding gRNA target sites, no annotation is available. (K) Molecular signature analysis of noncoding gRNA target sites, using adjacently located genes (one for each, 5' and 3'). 33 genes, out of the 211 genes analyzed, are part of the 'Biosynthetic process' group (which includes a total of 1,805 genes) with high confidence (p=3.4e-10).

DOI: https://doi.org/10.7554/eLife.42549.011

The following figure supplements are available for figure 5:

**Figure supplement 1.** Doxorubicin is toxic in hTERT-RPEI cells and TGW replicates correlate.

DOI: https://doi.org/10.7554/eLife.42549.012

**Figure supplement 2.** Quality control of the TGW validation library.

DOI: https://doi.org/10.7554/eLife.42549.013

the ASPA2 gene (chr2:9229295) (*Figure 5E* and *Supplementary file 12*). Within the noncoding gRNA target sites, we identified four gRNAs that targeted four different positions on chromosome X (56546543, 57766898, 63133046, 63245878), all of which are in close proximity to the SPIN2A gene (*Figure 5G* and *Supplementary file 13*), suggesting a doxorubicin-tolerance-inducing function in this locus.

In order to reveal whether the identified set of coding genes correlated with reported phenotypes or gene ontologies, we performed a molecular signature analysis of the 178 coding target regions and identified 25 genes that match the UV_RESPONSE_VIA_ERCC3 (downregulated in mutant ERCC3-expressing fibroblasts) group as the most significant hit (p-value of 6.11E-17) (*Figure 5H–I* and *Supplementary file 14*). Importantly, doxorubicin-induced interstrand crosslinks are repaired by ERCC3-dependent nucleotide excision repair (NER), and NER-deficient cells have been shown to display greater tolerance to adduct-forming anthracycline treatment, connecting these 25 genes to an increased doxorubicin tolerance (*Bret et al., 2013*; *Spencer et al., 2008*; *van Brabant et al., 2000*). Furthermore, mutations in noncoding sequences have been linked to the misregulation of adjacently located genes by disrupting cis-regulatory elements (*Hnisz et al., 2016*; *Katainen et al., 2015*; *Weinhold et al., 2014*). Therefore, we searched for available biotypes that are associated with the identified noncoding target regions and were able to identify target sites matching 'predicted promoter' (12.6%), 'lincRNAs' (11.3%) as well as 'processed pseudogenes' (4.5%) and 'CTCF binding sites' (3.6%) annotations (*Figure 5J* and *Supplementary file 13*). However, no biotype or genomic annotation was available for 52.7% of the noncoding gRNAs, and we therefore identified the nearest 5' and 3' located genes and used them to perform a molecular signature analysis (*Figure 5K* and *Supplementary file 15*). Among the four most enriched molecular signatures, we identified genes that are regulated by the transcription factors FOXO4, KLF1, and NFAT, noting that FOXO4 and NFAT downregulation has previously been reported to increase doxorubicin tolerance (*Figure 5K* and *Supplementary file 15*).

In summary, we explored the possibility of generating a partially degenerated SpCas9-gRNA library and its application in positive selection screens. Despite the limitations attributed to the generation of such a reagent and its applicability in cellular screens, we identified previously known and

unknown genes that are presumably linked to doxorubicin resistance. In addition, we identified non-coding sequence regions and their neighboring genes for which gene set enrichment analyses revealed an enrichment for transcription factors that are connected to increased doxorubicin tolerance.

## An optimized truly genome-wide 3Cs gRNA library

A library's sequence diversity and distribution directly dictates the experimental scale for positive and negative selection screens. Therefore, reducing the size of the TGW library to enable coverage-based screens is highly desirable. In line with this, gRNAs that are truncated to 17 nt have been demonstrated to maintain on-target efficiencies while reducing off-target effects (*Fu et al., 2014*; *Wyvekens et al., 2015*). We therefore truncated the degenerated TGW oligonucleotide sequence to 17 nt(optimized TGA, oTGW), approximating to a a total oligonucleotide diversity of $1.5 \times 10^9$ (*Figure 6A*). Importantly, the oTGW sequence diversity is 50-times smaller than the TGW sequence diversity, while the $1.65 \times 10^7$ unique target sequences in the human genome remain identical (*Figure 6A* and *Supplementary file 8*). As for the TGW oligonucleotide, we used hand-mixed phosphoramidite pools to synthesize the oTGW oligonucleotide and performed 3Cs reactions by combining this nucleotide with a ssDNA dU-template of the three conventionally used lentiviral CRISPR/Cas plasmids: pLentiGuide, pLentiCRISPRv2(Puro) and pLentiCRISPR(GFP-Puro). We then determined successful 3Cs reactions by gel electrophoresis (*Figure 6B*). Subsequent to bacterial amplification, an I-SceI clean-up step was performed before the three oTGW libraries were analyzed by NGS with an average of 28.4 million reads per library (*Figure 6C–E* and *Supplementary file 8*). Importantly, extracted gRNA-position nucleotide frequencies were extracted and translated to IUPAC nomenclature, revealing the initial oTGW degenerated oligonucleotide sequence (*Figure 6F–H*). Furthermore, an average wildtype remnant rate of 0.2% was determined and AUC values were 0.54 or below (*Figure 6—figure supplement 1*), suggesting a uniform distribution of represented gRNA sequences in all three oTGW libraries (*Makowski and Soares, 2003*). Thus, these oTGW libraries are the first of their kind, have the potential to elevate functional genomics approaches and will be made available to the scientific community by the Goethe University Depository (http://www.innovectis.de/INNOVECTIS-Frankfurt/Technologieangebote/Depository).

## Discussion

In the present study, we describe the 3Cs technology, an improved Kunkel mutagenesis protocol that facilitates the one-step and cloning-free generation of high-fidelity CRISPR/Cas and RNAi gene perturbation reagents. 3Cs uncouples sequence diversity from sequence distribution, making it useful for the generation of CRISPR/Cas gRNA libraries of arbitrary sequence diversities.

The 3Cs technology has several unique features. First, the bacteriophage-mediated generation of ssDNA makes the technology applicable to all plasmids containing a f1-origin of replication. Second, ssDNA-mutagenic oligonucleotides are annealed to the ssDNA of the template DNA, thereby circumventing the need for two oligonucleotides per gRNA and amplification by PCR, reducing associated costs and sequence bias. In line with this, T7 DNA polymerase, which is used in the 3Cs reaction, has an error rate of approximately $15 \times 10^{-6}$, resulting in as little as 0.0015% of mutated heteroduplex 3Cs product assuming 2 µg ssDNA of a 10 kb plasmid (*Kong et al., 1993*). Third, the presence of a gRNA placeholder sequence enables the near-complete removal of wildtype plasmid remnants. Last, we demonstrate 3Cs applicability and performance using the example of lentiviral plasmids. However, we foresee the plasmid range to be expanded to recombinant Adeno-Associated Virus (rAAV) plasmids and adenoviruses, as well as coding sequences for protein mutagenesis that enable in-cell and in-vivo functional screenings.

We demonstrate the fidelity and performance of 3Cs reagents by identifying the proliferative phenotype of human DUBs, validating previously known and uncovering hitherto unknown DUB phenotypes. We show that depletion of the DUBs USP28 and BRCC3 induces positive proliferation phenotypes, suggesting that they have tumor suppressive functions. In line with this, USP28 was recently identified as preventing p53 elevation in response to centrosome loss resulting from Plk4 inhibition, thereby preventing growth arrest in response to prolonged mitosis (*Meitinger et al., 2016*). On the other hand, we identify DUB enzymes whose depletion reduces cell fitness dramatically. Among them are COPS6 and USP7, both of which have been implicated in DNA damage response and

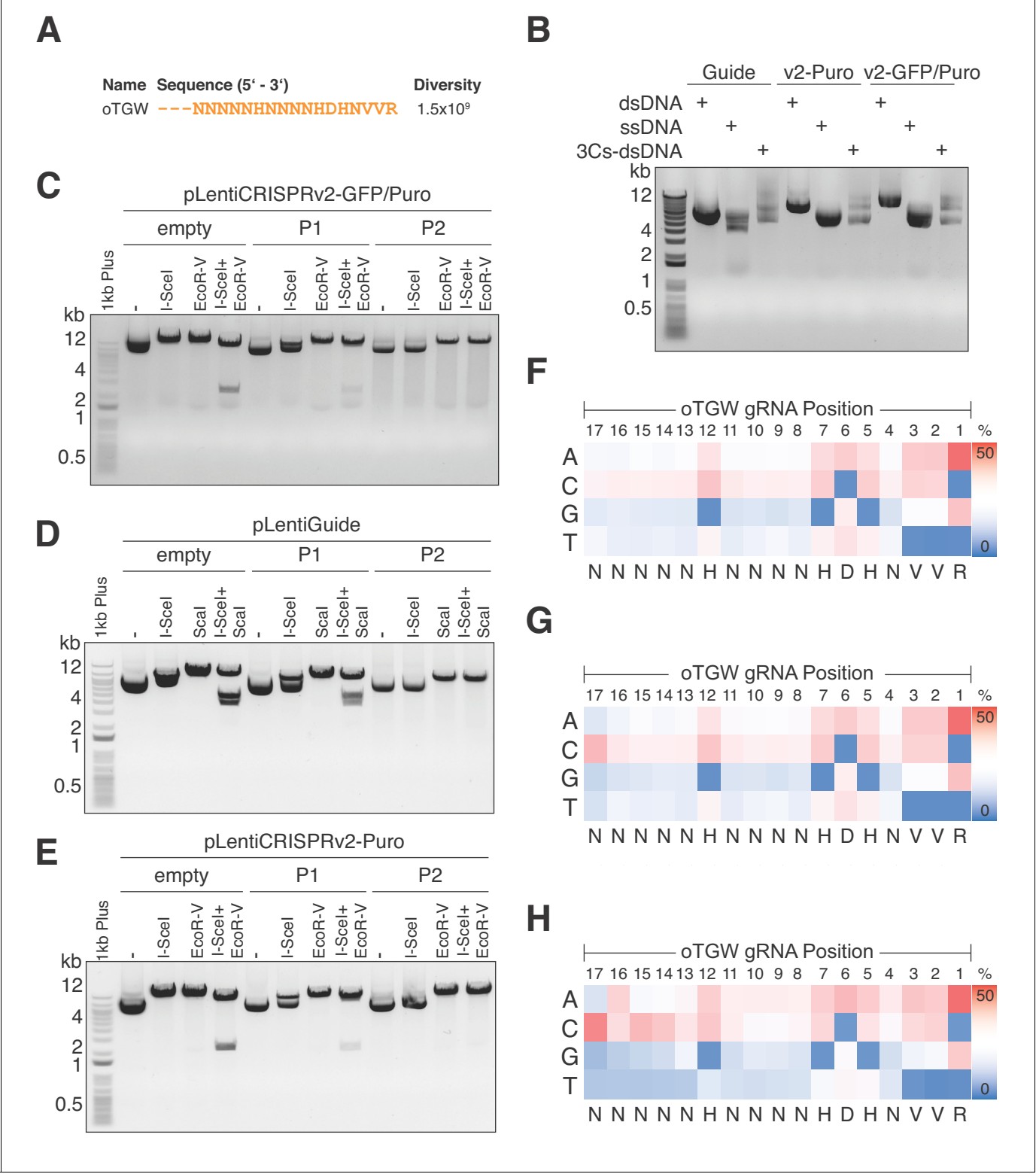

**Figure 6.** Optimized TGW (oTGW) libraries for functional interrogations in the coding and noncoding genome. (**A**) oTGW oligonucleotide sequence, based on reported SpCas9 nucleotide preferences. The truncation of three 5′ nucleotides results in 17-mer gRNAs with a total oligonucleotide diversity of $1.5 \times 10^9$. (**B**) oTGW 3Cs-dsDNA was synthesized on a ssDNA-template of pLentiGuide, pLentiCRISPRv2-Puro and pLentiCRISPRv2-GFP/Puro. 3Cs products are analyzed by gel electrophoresis on a 0.8% TAE/agarose gel. (**C–E**) Removal of template plasmid remnants with an I-SceI restriction enzyme digest. oTGW 3Cs-dsDNA was electroporated with efficiencies above $6.31 \times 10^9$ and amplified for DNA purification (P1). A subsequent I-SceI

*Figure 6 continued on next page*

*Figure 6 continued*

restriction enzyme digest and an electroporation of P1 yielded the final 3Cs libraries containing no detectable template plasmid (P2). An analytical restriction enzyme digest with I-SceI and EcoRV removes a 2.5-kb DNA fragment from the template plasmid (empty) and to a minor degree from P1 DNA pools. No 2.5-kb fragment could be observed in the final P2 DNA library pools, demonstrating the high purity of the final libraries (see also *Figure 6—figure supplement 1*). (F–H) High-throughput sequencing data derived from panels (C–E) were used to compute the nucleotide frequency of each gRNA nucleotide position, which are visualized as heat maps. The identified nucleotide frequencies closely resemble the pattern of the degenerated oTGW oligonucleotide shown in panel (A). Color coding illustrates the nucleotide frequencies (0% in blue to 50% in red).
DOI: https://doi.org/10.7554/eLife.42549.014

The following figure supplement is available for figure 6:

**Figure supplement 1.** oTGW quality control.
DOI: https://doi.org/10.7554/eLife.42549.015

enhanced p53 stability, leading to a prolonged $G_1$-phase cell cycle arrest (*Li et al., 2004*). In line with this, recent work demonstrates a direct connection between CRISPR/Cas gene editing and a p53-dependent DNA damage response that is associated with $G_1$ cell cycle arrest (*Haapaniemi et al., 2018*), suggesting that DNA damage-associated DUBs (e.g. COPS6 or USP7) can be used as p53 alternatives to control for DNA-damage-induced cell cycle arrest and negative hit calling in CRISPR/Cas functional genomic screens.

CRISPR/Cas functional genomic screens are widely used to interrogate protein-coding regions, but only few studies have used CRISPR/Cas gene editing to investigate the noncoding genome (*Canver et al., 2015*; *Diao et al., 2016*; *Korkmaz et al., 2016*; *Sanjana et al., 2016*). Although these studies have the potential to open a new area of functional genomics, their general applicability is limited to a predefined set of gRNAs and therefore to a small subset of genomic regions. A unique feature of the 3Cs technology is the uncoupling of gRNA sequence diversity from gRNA sequence distribution, facilitating the generation of partially randomized gRNA libraries. A fully randomized library with oligonucleotides of length 20 (20N) resembles the entire space of possible gRNA sequences and comprises $4^{20} = 1.1*10^{12}$ different sequences. Although an oligonucleotide pool covering this huge sequence space could theoretically be synthesized, there are at least two major reasons that render the experimental application unfeasible. The first reason is that the fraction of gRNA sequences that have a target site in the genome of interest would be very low. The ~300 million SpCas9 target sites in the human genome would be covered by a fully randomized library, but would represent only 0.027% of the library. Consequently, the second reason is that the experimental scale would have to be extremely high to cover all naturally occurring target sites, including all non-human targeting gRNAs that also need to be included, with sufficient coverage. Screening the 20N library with a coverage of 100 and a MOI of 0.5 would require $2.2*10^{14}$ cells, a cell number that is clearly not feasible in current experimental setups.

By focusing on SpCas9 nucleotide preferences, we introduce the partially randomized TGW library, which preferentially targets active gRNA sequences in the entire genome, including both coding and noncoding regions. The size of the TGW library is dramatically smaller than a fully randomized library but still comprises $7.3*10^{10}$ different gRNA sequences. We sought to explore the experimental application of such a large library and chose to screen for resistance to the cytotoxic agent doxorubicin in hTERT–RPE1 cells. Although insufficient TGW coverage led to low biological reproducibility, we were still able to retrieve gRNA overlap of three experiments. We identified protein-coding genes that have previously been associated with doxorubicin resistance such as CysLTR2, whose inhibition by two small chemical compounds reverted the doxorubicin-induced toxicity.

Interestingly, about half of the gRNAs for which we were able to identify a matching sequence in the human genome map to regions in the noncoding genome. Noncoding mutations have been shown to play pivotal roles in tumorigenesis by disrupting the function of cis-regulatory elements (e.g. promoters, enhancer, or transcription- factor binding sites) and topologically associating domains (TADs), thereby directly affecting the transcriptional regulation of adjacently located genes (*Katainen et al., 2015*; *Weinhold et al., 2014*). We annotated noncoding regions that are associated with hits from our TGW screen by mapping the corresponding gRNA sequences against the human reference genome. By allowing up to two mismatches, we attempted to account not only for exact matches but also for mismatched target sites. This approach yielded a number of

potential hits that should be interpreted with caution because cut sites might be called incorrectly and more stringent validation criteria are necessary. Nevertheless, we found a small set of gRNAs that presumably target predicted promoter sequences as well as CTCF and transcription factor binding sites, although further validation is necessary to gain mechanistic insights into how these sequences are linked to doxorubicin resistance.

Furthermore, our computational approach for coding and noncoding hit-calling is sensitive to incorrect calling of cut sites that can lead to false-positive target regions. We therefore suggest that the actual genome sequence of the cell line of interest is used in order to limit false-positive hit calling. We used the human reference genome (GRCh38.86), which might differ from the hTERT–RPE1 genome, potentially giving misleading conclusions in terms of hit calling. Another strategy to increase the rate of true-positive hits is to increase library coverage in the experiment. However, it is currently not feasible to screen either the 20N or the TGW library with sufficient coverage, at least not in adherent cell lines. To enable coverage-based truly genome-wide screenings, we reduced the TGW library diversity by truncating the library to 17-mers, yielding optimized TGW (oTGW) libraries that are more suited for high-throughput experiments with suspension cells. We propose that the oTGW CRISPR/Cas gRNA libraries are suitable for broad biological screenings with the highest genetic target complexity. Such screens can be followed by targeted validation screens using newly synthesized libraries that are tailored to the initially identified cut sites, which will enable rapid functional validation of such complex genetic experiments.

# Materials and methods

**Key resources table**

| Reagent type (species) or resource | Designation | Source or reference | RRID identifiers |
|---|---|---|---|
| Cell line (human) | HEK293T | ATCC | RRID:CVCL_0063 |
| Cell line (human) | hTERT-RPE1 | ATCC | RRID:CVCL_4388 |
| Cell line (human) | RPE1 | Ian Cheeseman | |
| Antibody | Anti-GFP (B-2) | Santa Cruz Biotechnology | RRID:AB_627695 |
| Antibody | Anti-alpha Tubulin | DSHB | RRID:AB_2315509 |
| Antibody | Goat anti-Mouse IgG (H + L) Secondary Antibody | Thermo Fisher Scientific | RRID:AB_228307 |
| Antibody | Goat anti-Rabbit IgG (H + L) Secondary Antibody | Thermo Fisher Scientific | RRID:AB_228341 |
| Bacteria (*E. coli*) | K12 CJ236 | NEB (E4141) | |
| Bacteria (*E. coli*) | 10 beta | NEB (C3020K) | |
| Recombinant DNA reagent | pLentiGuide | Addgene (52963) | |
| Recombinant DNA reagent | pLentiCRISPRv2 | Addgene (52961) | |
| Recombinant DNA reagent | PLKO.1 | Addgene (8453) | |
| Recombinant DNA reagent | pPax2 | Addgene (12260) | |
| Recombinant DNA reagent | pMD2.G | Addgene (12259) | |
| Commercial kit | E.Z.N.A. M13 DNA Mini Kit | Omega Bio-Tek (D69001-01) | |
| Commercial kit | GeneJET Gel extraction kit | Thermo Fisher (K0692) | |

*Continued on next page*

*Continued*

| Reagent type (species) or resource | Designation | Source or reference | RRID identifiers |
|---|---|---|---|
| Commercial kit | Plasmid Maxi Kit | Qiagen (12163) | |
| Commercial kit | PureLink Genomic DNA Mini Kit | Invitrogen (K1820-01) | |
| Chemical compound, drug | Ampicillin | Roth (K029.2) | |
| Chemical compound, drug | Chloramphenicol | Roth (3886.1) | |
| Chemical compound, drug | Kanamycin | Roth (T832.3) | |
| Chemical compound, drug | NaCl | Roth (31434) | |
| Chemical compound, drug | ATP | NEB (756) | |
| Chemical compound, drug | DTT | Cell Signaling Technology Europe (7016) | |
| Chemical compound, drug | dNTP mix | Roth (0178.1/2) | |
| Chemical compound, drug | Penicillin-streptomycin | Sigma-Aldrich (P4333) | |
| Chemical compound, drug | Hygromycin | Capricorn Scientific (HYG-H) | |
| Chemical compound, drug | Lipofectamin 2000 | Thermo Fisher (11668019) | |
| Chemical compound, drug | Polybrene | Sigma Aldrich (H9268) | |
| Chemical compound, drug | Doxorubicin | Selleckchem (S1208) | |
| Chemical compound, drug | Bay-CysLT2 | Cayman Chemical (10532) | |
| Chemical compound, drug | Bay-u9773 | Tocris Bioscience (3138) | |
| Other | T4 DNA ligase | NEB (M0202) | |
| Other | T7 DNA polymerase (unmodified) | NEB (M0274) | |
| Other | 2 mm electroporation cuvette | BTX (45–0125) | |
| Other | Gene Pulser electroporation system | BioRad (164–2076) | |
| Other | I-SceI | NEB (R0694) | |
| Other | DMEM | Thermo Fisher (41965–039) | |
| Other | DMEM/F12 | Thermo Fisher (11320–074) | |
| Other | FBS | Thermo Fisher (10270) | |
| Other | M13KO7 helper phage | NEB (N0315) | |
| Other | Polyethylene glycol | Roth (263.2) | |
| Other | SOC outgrowth medium | Thermo Fisher (15544034) | |
| Other | 2YT medium | Roth (6676.2) | |
| Other | T4 polynucleotide kinase | NEB (M0201) | |

*Continued on next page*

*Continued*

| Reagent type (species) or resource | Designation | Source or reference | RRID identifiers |
|---|---|---|---|
| Other | T7 endonuclease | NEB (M0302) | |
| Other | OneTaq DNA polymerase | NEB (M0480) | |
| Other | Next High-Fidelity 2x PCR Master Mix | NEB (M0541) | |
| Other | NextSeq 500 | Illumina | |
| Software, algorithm | bcl2fastq | Illumina | RRID:SCR_015058 |
| Software, algorithm | cutadapt v.1.15 | *Martin, 2011* | RRID:SCR_011841 |
| Software, algorithm | CasOFFinder v2.4 | *Bae et al., 2014* | |
| Software, algorithm | SnpEff 4.3T | *Cingolani et al., 2012* | RRID:SCR_005191 |
| Software, algorithm | MAGeCK | *Li et al., 2014* | |

## Cloning of 3Cs template plasmids

The NHT and I-SceI gRNA sequences (see 'DNA oligonucleotides') were annealed and cloned into pLentiGuide (Addgene 52963) and pLentiCRISPRv2 (Addgene 52961) via BsmBI restriction enzyme digest (NEB, R0580) and subsequent ligation with T4 ligase (NEB, M0202). Correct clones were identified by Sanger sequencing at Microsynth SeqLab, Switzerland, using a U6 primer (see 'DNA oligonucleotides').

## 3Cs oligonucleotide design

All of the 3Cs oligonucleotides that were used in experiments are listed in 'DNA oligonucleotides'. DNA oligonucleotides were purchased from Sigma-Aldrich and Integrated DNA Technologies (IDT) as single or pooled oligonucleotides, and from Twist Bioscience or CustomArray Inc. as oligonucleotide pools. The 3Cs oligonucleotides were designed with two homology regions flanking the intended 20-nt gRNA sequence. The homology regions were at least 15 nt in length ($T_m$ above 50°C) and matched the 3′ end of the U6 promoter region and the 5′ start of the gRNA scaffold in the template plasmids. The TGW and the oTGW 3Cs oligonucleotides were designed on the basis of a pattern of nucleotide preferences as previously determined (*Doench et al., 2014*; *Doench et al., 2016*). The observed nucleotide preferences were translated into a degenerated 17-nt DNA sequence (oTGW, see 'DNA oligonucleotides'). The randomized oligonucleotides for the six libraries of increasing diversity each had stretches of an increasing number of fully randomized nucleotides (see 'DNA oligonucleotides'). The oligonucleotide with four randomized positions was designed to contain the stretch of four consecutive Ns beginning at position 30 of the oligonucleotide. Oligonucleotides with increasing randomization were designed by extending the randomized pattern in an alternating fashion left and right by one randomized position each. The randomized segments and the flanking constant regions were designed to replace the I-SceI recognition site in the template plasmid to enable the clean-up digestion step. In general, every gRNA was designed to avoid the occurrence of the I-SceI recognition site.

## Overview of reagents and equipment needed for the synthesis of 3Cs

### Equipment

Desktop microcentrifuge, shaking incubator at 37°C, 1.5 ml collection tubes, filtered sterile pipette tips, thermoblocks at 90°C and 50°C (e.g., Thermo Fisher, 88870004), an ultracentrifuge capable of spinning 50 ml falcon tubes at 10,000 rpm (Beckman Coulter Avanti J-30 I ultracentifuge and a Beckman JA-12 fixed angle rotor), falcon tubes (polypropylene, 50 ml (Corning 352070)), a Bio-Rad Gene Pulser electroporation system (BioRad 164–2076), electroporation cuvettes Plus (2 mm, Model no. 620 (BTX)), a gel electrophoresis chamber, and erlenmeyer flasks (glass, 100 ml).

### KCM Transformation

5x KCM buffer (0.5M KCl, 0.15M CaCl$_2$, 0.25M MgCl$_2$), *Escherichia coli* strain K12 CJ236 (NEB, E4141), SOC outgrowth medium (ThermoFisher Scientific, 15544034), LB-agar plates supplemented with 100 µg/ml ampicillin (Roth, K029.2).

### Phage amplification and ssDNA purification

2YT media (Roth, 6676.2), M13KO7 helper phage (NEB, N0315), ampicillin (Roth, K029.2), chloramphenicol (Roth, 3886.1), kanamycin (Roth, T832.3), uridine (Sigma-Aldrich, U3750), 20% PEG/NaCl (20% polyethylene glycol (Roth, 0263.2), 2.5 M NaCl (Roth, 31434)), Dulbecco's phosphate-buffered saline (PBS, Sigma, D8662), E.Z.N.A. M13 DNA Mini Kit (Omega Bio-Tek, D69001-01), store purified phage in PBS at 4°C.

### 3Cs-DNA synthesis

10x TM buffer (0.1 M MgCl$_2$, 0.5 M Tris-HCl, pH 7.5), 10 mM ATP (NEB, 0756), 100 mM DTT (Cell Signaling Technology Europe, 7016), T4 polynucleotide kinase (NEB, M0201), 100 mM dNTP mix (Roth, 0178.1/2), T4 DNA ligase (NEB, M0202), T7 DNA polymerase (unmodified) (NEB, M0274), Thermo Fisher Scientific GeneJET Gel Extraction Kit (Thermo Fisher, K0692), 3M sodium acetate (Sigma-Aldrich, 71196).

### Electroporation and I-SceI clean-up digest

2 mm cuvette (BTX, 45–0125), electrocompetent *E. coli* (10-beta, NEB, C3020K), SOC outgrowth medium (Thermo Fisher, 15544034), LB-media (Roth, X964.3) supplemented with 100 µg/ml ampicillin, Qiagen Plasmid Maxi Kit (Qiagen, 12163), I-SceI (NEB, R0694), NEB CutSmart buffer (NEB, B7204), 0.5% TAE/agarose gel, Thermo Fisher Scientific GeneJET Gel Extraction Kit.

## dU-ssDNA template amplification

Bacteria (*Escherichia coli* strain K12 CJ236, NEB, E4141) were transformed with 500 ng of template plasmid according to the following protocol: DNA was mixed with 2 µl of 5x KCM buffer (0.5M KCl, 0.15M CaCl$_2$, 0.25M MgCl$_2$) set to 10 µl with water and chilled on ice for 10 min. An equal volume of CJ236 bacteria was added to the DNA/KCM mixture, gently mixed, and incubated on ice for 15 min. The bacteria–DNA mixture was then incubated at room temperature for 10 min, and subsequently inoculated into 200 µl of prewarmed SOC media (ThermoFisher Scientific, 15544034). Bacteria were incubated at 37°C and 200 rpm for 1 hr and then selected with ampicillin (100 µg/ml) on LB-agar plates overnight at 37°C.

The next morning, a single colony of transformed CJ236 was picked into 1 ml of 2YT media (Roth, 6676.2) supplemented with M13KO7 helper phage (NEB, N0315) to a final concentration of 1 $\times$ 10$^8$ pfu/ml, chloramphenicol (final concentration 35 µg/ml), and ampicillin (final concentration 100 µg/ml) to maintain the host F′ episome and the phagemid, respectively. Supplementation of uridine (Sigma-Aldrich, U3750) was set to 2.5 µM. After 2 hr of shaking at 200 rpm and 37°C, kanamycin was added to a final concentration of 25 µg/ml to select for bacteria that have been infected with M13KO7 helper phage. Bacteria were kept at 200 rpm and 37°C for an additional 6 hr before the culture was transferred to 30 ml of 2YT media supplemented with ampicillin (final concentration 100 µg/ml) and kanamycin (final concentration 25 µg/ml). After 20 hr of shaking at 200 rpm and 37°C, the bacterial culture was centrifuged for 10 min at 10,000 rpm and 4°C in a Beckman JA-12 fixed angle rotor. To precipitate phage particles, the supernatant was transferred to 6 ml (1/5 of culture volume) PEG/NaCl (20% polyethylene glycol 8,000, 2.5 M NaCl), incubated for 1 hr at room temperature and subsequently centrifuged for 10 min at 10,000 rpm and 4°C in a Beckman JA-12 fixed angle rotor. The phage pellet was resuspended in 1 ml Dulbecco's phosphate-buffered saline (PBS, Sigma, D8662) and centrifuged at 13,000 rpm for 5 min, before the phage-containing supernatant was stored at 4°C. Circular ssDNA was purified from the resuspended phages with the E.Z.N.A. M13 DNA Mini Kit (Omega Bio-Tek, D69001-01) according to the manufacturer's protocol, and purified ssDNA was stored at 4°C.

## 3Cs-DNA synthesis

The oligonucleotides that were used for 3Cs reactions and the suppliers are listed separately (see 'DNA oligonucleotides'). 3Cs oligonucleotides for specific pools were mixed in equimolar ratios. 600

ng of pooled oligonucleotides were phosphorylated by mixing them with 2 µl 10x TM buffer (0.1 M MgCl$_2$, 0.5 M Tris-HCl, pH 7.5), 2 µl 10 mM ATP (NEB, 0756), 1 µl 100 mM DTT (Cell Signaling Technology Europe, 7016), 20 units of T4 polynucleotide kinase (NEB, M0201) and water to a total volume of 20 µl. The mixture was incubated for 1 hr at 37°C.

Phosphorylated oligonucleotides were annealed to the circular dU-ssDNA template by adding 20 µl of phosphorylation product to 25 µl 10x TM buffer, 20 µg of dU-ssDNA template, and water to a total volume of 250 µl. The mixture was denatured for 3 min at 90°C, annealed for 5 min at 50°C, and cooled down for 5 min at room temperature.

3Cs-DNA was generated by adding 10 µl of 10 mM ATP, 10 µl of 100 mM dNTP mix (Roth, 0178.1/2), 15 µl of 100 mM DTT, 2000 ligation units of T4 DNA ligase (NEB, M0202), and 30 units of T7 DNA polymerase (NEB, M0274) to the annealed oligonucleotide–ssDNA mixture. The 3Cs synthesis mix was incubated for 12 hr (overnight) at room temperature. The 3Cs synthesis product was then affinity purified and desalted using a Thermo Fisher Scientific GeneJET Gel Extraction Kit (Thermo Fisher, K0692) according to the following protocol: 600 µl of binding buffer and 5 µl 3M sodium acetate (Sigma-Aldrich, 71196) were added to the synthesis product, mixed and applied to two purification columns, which were centrifuged for 3 min at 460 g (2,500 rpm in a Sigma-Aldrich 1–14 table top centrifuge). The flow-through was applied a second time to the same purification column to maximize yield and centrifuged for 3 min at 460 g. DNA was eluted in 40 µl warm water. The 3Cs reaction product was analyzed by gel electrophoresis alongside the dU-ssDNA template on a 0.8% TAE/agarose gel (100 V, 30 min). 3Cs-shRNA libraries were synthesized according to the protocol described above with the following modifications: in two setups, either 60 ng or 120 ng of circular template dU-ssDNA of pLKO.1 (Addgene: 1864) was used.

## Electroporations and I-SceI clean-up digest

To generate pool 1 (P1) of a library, 3Cs-DNA constructs were electroporated with a cold 2 mm cuvette (BTX, 45–0125) into electrocompetent *E. coli* (10-beta, NEB, C3020K) using a Bio-Rad Gene Pulser with the following settings: resistance 200 Ohm, capacity 25 F, voltage 2.5 kV. 2 µg of DNA was mixed with 400 µl of freshly thawed cells. Electroporated cells were rescued in 25 ml of pre-warmed SOC media and incubated for 30 min at 37°C and 200 rpm.

After 30 min of incubation, a dilution series was performed to determine the transformation efficiency and the number of transformed bacteria. 10 µl of culture was diluted $10^{-1}$ to $10^{-12}$ and plated on LB agar plates supplemented with 100 µg/ml ampicillin. The remaining culture was added to 200 ml LB-media (Roth, X964.3) supplemented with 100 µg/ml ampicillin. Plates were incubated overnight at 37°C, the liquid cultures were incubated overnight at 37°C and 200 rpm. The next day, the electroporation efficiency and the number of transformed bacteria were determined. The plasmid DNA of the overnight liquid cultures was purified using a Qiagen Plasmid Maxi Kit (Qiagen, 12163) according to the manufacturer's protocol.

To generate the final pool 2 (P2) of a library, 10 µg of purified P1 DNA was digested with 50 units I-SceI (NEB, R0694) and 5 µl NEB CutSmart buffer (NEB, B7204) in a reaction volume of 50 µl for 1.5 hr at 37°C. The digestion reaction was subjected to gel electrophoresis on a 0.5% TAE/agarose gel (100 V, 30 min) to separate the undigested 3Cs synthesis product from linearized template plasmid. The band resembling the undigested correct 3Cs synthesis product was purified using a Thermo Fisher Scientific GeneJET Gel Extraction Kit according to the manufacturer's protocol. In a second step, the purified 3Cs synthesis product was electroporated according to the electroporation protocol described above. The final P2 library preparation was purified from liquid culture using a Qiagen Plasmid Maxi Kit according to the manufacturer's protocol and quality controlled with analytical restriction enzyme digests. 3Cs-shRNA pools were generated according to the above protocol with the following modifications. Instead of using I-SceI for the clean-up digestion, we used Bsu36I to digest template plasmid remnants in the first DNA pool (P1). P1 was electroporated using the settings described above to yield the final pool (P2). Both 3Cs-shRNA pools were purified from liquid culture using a Qiagen Plasmid Maxi Kit according to the manufacturer's protocol and were quality controlled with analytical restriction enzyme digests and Sanger sequencing.

## Cell culture

HEK293T cells (ATCC, CRL-3216) were maintained in Dulbecco's Modified Eagle's Medium (DMEM, Thermo Fisher Scientific, 41965–039) and hTERT–RPE1 cells (ATCC, CRL-4000 and Ian Cheeseman's) in DMEM: Nutrient Mixture F-12 (DMEM/F12, Thermo Fisher Scientific, 11320–074), each supplemented with 10% fetal bovine serum (FBS, Thermo Fisher Scientific, 10270) and 1% penicillin-streptomycin (Sigma-Aldrich, P4333) at 37°C with 5% $CO_2$. In addition, hTERT–RPE1 cells were supplemented with 0.01 mg/ml hygromycin B (Capricorn Scientific, HYG-H). hTERT–RPE1 cells were obtained from ATCC/LGC (CRL-4000) and Ian Cheeseman. No method to ensure the state of authentication has been applied. Mycoplasma contamination testing was performed immediately after the arrival of the cells and multiple times during the course of the experiments.

## Cell extracts and antibodies

Preparation of lysates and immunoblot analyses were performed as described previously using Tris lysis buffer (50 mM Tris–HCl (pH 7.8), 150 mM NaCl, 1% IGEPAL CA-630) containing 20 mM NaF, 20 mM β-glycerophosphate, 0.3 mM Na-vanadate, 20 μg/ml RNase A, 20 μg/ml DNase and 1/300 protease inhibitor cocktail (Sigma-Aldrich, P8340) and phosphatase inhibitor cocktail #2 (Sigma-Aldrich, P5726) (*Kaulich et al., 2015*). The antibodies used in this study were purchased from the following sources: mouse anti-GFP (GFP (B-2): sc-9996, 1:2,000, Santa Cruz Biotechnology, Inc.), mouse anti-Tubulin (clone 12G10, 1:1,000, Developmental Studies Hybridoma Bank, University of Iowa). Secondary antibodies used for western blot analysis were goat anti-mouse (Thermo Scientific, 31430) and goat anti-rabbit (Thermo Scientific, 31460). The mouse anti-Tubulin hybridoma cell line (clone #12G10) was developed by J. Frankel and E.M. Nelson under the auspices of the NICHD and maintained by the Developmental Studies Hybridoma Bank. Protein levels were visualized with Pierce ECL Western Blotting Substrate on a BioRad ChemiDoc MP imaging system and analyzed with Bio-Rad Image Lab software (version 4.1 build 16).

## Generation and quantification of lentiviral particles

The day before transfection, HEK293T cells were seeded to $5 \times 10^5$ cells/ml. To transfect HEK293T cells, transfection media containing 1/10 of culture volume Opti-MEM I (Thermo Fisher Scientific, 31985–047), 10.5 μl Lipofectamin 2000 (Thermo Fisher Scientific, 11668019), 1.65 μg/ml transfer vector, 1.35 μg/ml pPAX2 (Addgene, 12260) and 0.5/ml μg pMD2.G (Addgene, 12259) was prepared. The mixture was incubated for 30 min at room temperature and added drop-wise to the media. The next morning, the transfection medium was replaced with fresh media to remove the transfection reagent. Lentiviral supernatant was harvested at 24 hr and 48 hr after transfection, pooled and stored at −80°C.

To determine the lentiviral titer, hTERT–RPE1 cells were plated in a 24-well plate with 20,000 cells per well. The following day, cells were transduced using 8 μg/ml polybrene (Sigma, H9268) and a series of 0.5, 1, 5, and 10 μl of viral supernatant. After 3 days of incubation at 37°C, the percentage of fluorescence-positive cells was determined by flow cytometry. The following formula was used to calculate the viral titer: $Virus titer\ (transducing\ units/mL) = \frac{20.000\ target\ cells \times \frac{\%\ of\ GFP\ positive\ cells}{100}}{volume\ of\ supernatant\ (mL)}$.

Alternatively, lentiviral titers were determined by colony formation titering assay for lentivirus.

## Flow cytometry

All samples were analyzed on a FACSCanto II flow cytometer (BD Biosciences), and data were processed by FlowJo software (FlowJo, LLC). Gating was carried out on the basis of viable and single cells that were identified on the basis of their scatter morphology.

## Lentiviral transduction

hTERT–RPE1 cells were seeded at an appropriate density for each experiment with a maximal confluency of 60–70% in DMEM/F12, supplemented with 10% FBS, 0.02 μg/ml hygromycin, and 1% penicillin-streptomycin. On the day of transduction, polybrene was added to the media to a final concentration of 8 μg/ml. The volume of lentiviral supernatant was calculated on the basis of the diversity of the respective library and of the desired coverage and multiplicity of infection (MOI) of the experiment. The number of cells that were transduced at the beginning of an experiment was

calculated by multiplying the diversity of the library with the desired coverage and the desired MOI. For example, the parameters for the DUB library screen were set at a coverage of 1.000 and an MOI of 0.2, that is one lentiviral particle per five cells. The total number of cells that were transduced was calculated as follows: 363 * 1,000 * 5 = 1,815,000. The next morning, the medium was replaced with fresh media and the cells were subjected to antibiotic selection or experimental analysis.

## Homology arm lengths and 3Cs reaction times

To test different homology arm lengths, four 3Cs reactions were performed using four different oligonucleotides with increasing lengths of homology to the pLentiGuide NHT, according to the 3Cs synthesis protocol described above. The reaction products were analyzed by gel electrophoresis.

To monitor the 3Cs synthesis process over time, we annealed the TGW oligonucleotide to the pLentiGuide NHT and generated 3Cs-dsDNA. 2 µl of the reaction was sampled from the reaction tube and transferred to −20°C at different timepoints from 0 hr to 20 hr. All samples were analyzed together by agarose gel electrophoresis. To visualize the kinetics of the 3Cs reactions, 3Cs-dsDNA band intensities were determined and normalized to time point 0 before plotting against the time of their harvest using the Bio-Rad Image Lab software (version 4.1 build 16).

## eGFP gene editing and T7 endonuclease I assay

The efficiency of eGFP gene editing was analyzed by transducing eGFP-expressing hTERT–RPE1 cells with 3Cs gRNA constructs based on pLentiCRISPRv2, a subsequent T7 Endonuclease I assay, and immunoblotting. The experiment was performed in triplicates using a control gRNA (NHT), a single GFP-targeting 3Cs-gRNA (GFP#1) or a pool of six GFP-targeting 3Cs-gRNAs (GFP#1–6). After 7 d of incubation at 37°C without antibiotic selection, cells were trypsinized and the genomic DNA was purified using a PureLink Genomic DNA Mini Kit (Invitrogen, K1820-01) according to the manufacturer's protocol.

To assess the genome targeting efficiency of the 3Cs reagents, we analyzed the four cell populations that were transduced with the NHT-gRNA, the GFP#1 gRNA or the GFP#1–6 pool, or that were not transduced at all. We PCR-amplified the GFP locus with OneTaq DNA polymerase (NEB, M0480) using 1 µg of genomic DNA, 40 µM dNTPs (final concentration), 0.2 µM of each forward and reverse amplification primer (see 'DNA oligonucleotides: eGFP T7 forward and eGFP T7 reverse'), 10x OneTaq standard buffer, and 2.5 units of OneTaq DNA polymerase. The cycles were set up as follows: initial denaturation at 94°C for 3 min, 39 cycles of denaturation at 94°C for 20 s, annealing at 55°C for 30 s, strand extension at 68°C for 2 min, and final strand extension at 68°C for 5 min. The PCR products were analyzed on a 0.8% TAE/agarose gel (100 V, 30 min) and purified using a Thermo Fisher Scientific GeneJET Gel Extraction Kit according to the manufacturer's protocol. The T7 endonuclease I digestion was assembled with 6 µg of purified PCR product, 10x NEBuffer 2 water to 48 µl, denatured at 95°C for 5 min, and annealed in two steps from 95–85°C with −2 °C/second, and from 85–25°C with −0.1 °C/second. To the annealed PCR product, 7 µl of T7 Endonuclease I (NEB, M0302) was added and incubated for 15 min at 37°C. The fragmented PCR products were analyzed on a 0.8% TAE/agarose gel (100 V, 30 min) and band intensities were determined using the Bio-Rad Image Lab software (version 4.1 build 16).

## DUB proliferation screen

The DUB proliferation screen was performed in biological duplicates. hTERT–RPE1 cells were transduced with lentiviral supernatant with a MOI of 0.2 and a library coverage of 1,000. For each replicate and time point, 2.5 million cells were seeded. Cells corresponding to the control time point were harvested 2 d post-transduction. All remaining cells were kept in growing and library-diversity-maintaining conditions in the presence of 10 µg/ml puromycin. After 11 d and 21 d, cells were harvested and their genomic DNA purified and processed for NGS. Validation of DUB screen hit candidates was performed in hTERT–RPE1 cells with 3Cs-shRNA-mediated target gene knockdown and the subsequent assessment of cell proliferation used an AlamarBlue assay (Bio-Rad, BUF012A).

Doxorubicin-resistance-screen hTERT–RPE1 cells were treated with increasing concentrations of doxorubicin, ranging from 0 to 1,000 nM, for four consecutive days. After 4 d, the treatment was stopped by changing the medium to doxorubicin-free medium and cells were cultivated for another

4 d. After a total of 8 d, cell viability was determined and quantified with an AlamarBlue assay (Bio-Rad, BUF012A).

To screen for doxorubicin resistance, the TGW library was delivered in triplicates to a total of 5.5 × 10⁸ hTERT–RPE1 cells with doxycycline-inducible Cas9 expression via lentiviral transduction at a MOI of 1. Transduced cells were cultured for 7 d in standard medium supplemented with 1 µM doxycycline (Sigma-Aldrich) and 10 µg/ml puromycin. At day 7, the medium was changed to selection medium containing 1 µM doxorubicin (Selleckchem, S1208). After 3 wk of selection (fresh doxorubicin every 4 d), surviving cells were harvested and processed for NGS.

## NGS of plasmid and genomic DNA

To purify genomic DNA, surviving cells were trypsinized and pelleted. Genomic DNA was extracted using the PureLink Genomic DNA Mini Kit according to the manufacturer's protocol. For NGS library preparation, 100 ng of plasmid or up to 2 µg of genomic DNA per reaction was used in a 50 µl PCR reaction using Next High-Fidelity 2x PCR Master Mix (NEB, M0541) (according to the manufacturer's protocol) and 1 µl of 10 µM primers each of forward and reverse primers. Primer sequences are listed separately (see 'DNA oligonucleotides'). The sequencing primers contained an 8-nt long barcode sequence, enabling the multiplexing of several samples in a single sequencing run and Illumina adapter sequences. Thermal cycler parameters were set as follows: initial denaturation at 98°C for 5 min, 19 cycles of denaturation at 98°C for 30 s, annealing at 55°C for 30 s, extension at 72°C for 1 min, and final extension at 72°C for 5 min. PCR products were purified from a 0.5% TAE/agarose gel using a Thermo Fisher Scientific GeneJet Gel Extraction Kit according to the manufacturer's protocol. The purified PCR product was prepared to a final concentration of 2.4 pM in a total volume of 2.2 ml and loaded onto a NextSeq 500 sequencer (Illumina), according to the manufacturer's protocol. Sequencing was performed with single end reads, 75 cycles and 8 cycles of single index reading.

## Data processing and analysis

All data obtained from NGS were demultiplexed using the Illumina command line tool bcl2fastq, v2.17. gRNA representation of all libraries was assessed using cutadapt v1.15 (*Martin, 2011*) and custom Python scripts. In brief, 3′ sequencing adapters were trimmed using a prefix of the 3′ homology sequence; trimmed reads were further trimmed by keeping only the last 20 nucleotides for all libraries except the oTGW, for which the last 17 nt were kept. Only reads with no ambiguously sequenced nucleotides were considered for further analyses. For the TGW and the oTGW, the resulting sequences were compared to the TGW or oTGW DNA sequence pattern, respectively, using Python and regular expressions. The reads obtained from sequencing the six randomized libraries with diversities ranging from 256 to 262,144 gRNAs were processed similarly by comparing the trimmed reads with the gRNA pattern of the respective library. For the GFP and DUB libraries, the reads were aligned to the respective sequence library. Matching sequences were counted to determine the read count distribution of a sample. The read counts of individual gRNAs for a sample were normalized by the total number of read counts that could be assigned to the respective library. The screening of samples after treatment of the cells was carried out in the same way. To determine the dispersion of the read counts, the coefficient of variation was computed by dividing the standard deviation of the normalized read counts by the mean of the normalized read count $\bar{x}$, $CV = \frac{s}{\bar{x}}$. To assess the uniformity of each library distribution, we generated Lorenz curves of gRNA representation. The Lorenz curves of gRNA representation rank gRNAs by abundance scaled to 1 and show the fraction of total sequencing reads that are represented by the sum of gRNA read counts. The area under the curve (AUC) was computed in GraphPad Prism 5.0b for Mac (GraphPad Software, La Jolla California USA, www.graphpad.com) or with a custom Python script using Numpy 1.14.2 (*Oliphant, 2010*). Heat maps were generated by accumulating the nucleotide frequency at each position of the sequenced reads and normalized by the total number of read counts.

To correct the read counts of the six randomized libraries with diversities ranging between 256 and 262,144 gRNAs for C bias, we determined the nucleotide frequencies for each sequence position of the trimmed and final reads and normalized the observed frequencies to the expected nucleotide frequency of 25%. Each read was then scored by summing the normalized frequencies for all reads individually. The observed read count per gRNA was then multiplied with this score, divided by the sum of all read counts that matched the respective gRNA pattern, and normalized to the sum

of all corrected and normalized read counts. Lorenz curves were generated on the basis of the corrected and normalized read counts.

Read count data from the DUB screen were analyzed by summing all individual gRNA read counts per gene and normalizing each gene read count per sample to the total number of read counts within that sample. Spearman correlation and Shapiro-Wilk confidence tests were performed to assess the reproducibility of the DUB screen replicates. MAGeCK and PinAPL-Py were used to analyze the read counts of both replicates and to calculate aggregated positive and negative proliferation phenotypes by means of $\log_2$ fold changes with associated p-values.

## Analysis of gRNA on- and off-target locations

To determine the on- and off-targets of the 4232 hits from the doxorubicin resistance screen, Cas-OFFinder (v2.4) was applied to search the human genome (GRCh38.86) for gRNA target sites with up to two mismatches (*Bae et al., 2014*). The genomic positions of each on- and off-target were annotated with Ensembl genome assembly GRCh38.86, using SnpEff 4.3T (*Cingolani et al., 2012*) and custom Python scripts. Multiple annotations for a location were collapsed onto a single gene type and the corresponding gene name, if available. Genomic locations associated with an intergenic region were not considered to be annotated. Additional noncoding, regulatory, and pseudogene information was annotated using the Ensembl regulatory and motif features from release 91 and the Gencode consensus pseudogenes dataset from release 27 (GRCh38.p10). Additional standard annotation data from Gencode, release 27, were also included. Spearman rank and Pearson correlation were computed with NumPy (1.14.2). To determine the putative effect of gRNA off-targets on previously identified on-target locations, we mapped the gene names that were associated with off-targets back to the genes that were associated with on-target hits.

## Validation of TGW doxorubicin-resistance screening hits

To validate CysLTR2 as a doxorubicin-resistance-inducing gene, we applied the CysLT2 receptor antagonists Bay-CysLT2 (Cayman Chemical, 10532) and Bay-u9773 (Tocris Bioscience, 3138). In two different triplicates, cells were treated with with increasing concentrations (0 µM, 0.01 µM, 0.05 µM, 0.1 µM, 0.25 µM, 0.5 µM, 0.5 µM, 1 µM) of each inhibitor as well as with increasing doxorubicin concentrations (0 µM, 1 µM, and 10 µM). After 4 d, cell survival was assessed with an AlamarBlue assay. 10% AlamarBlue was added to the cultured cells and incubated for 2 hr at 37°C, and fluorescence was measured with an excitation wavelength of 560 nm and a fluorescence emission of 590 nm on a BioTek Synergy H1 microplate reader. The given measured fluorescence emissions were averaged over all replications for each experiment.

The 4,232 hits that were found in the TGW doxorubicin screen were compiled into an individual 3Cs library (validation library). The validation library was generated according to the 3Cs DNA synthesis protocol described above. We seeded $1 \times 10^6$ hTERT–RPE1 in T175 cell culture flasks in DMEM/F12 and transduced them the next day with lentivirus of the validation library using 8 µg/ml polybrene with an MOI of 0.1 and an experimental coverage of 1,000. After 3 d, the control cells were harvested. The screen was conducted with 2.5 µg/ml (final concentration) puromycin selection and 1 µM doxorubicin treatment. The medium was changed every third day to maintain constant puromycin and doxorubicin concentrations. After three weeks of selection, surviving cells were harvested and processed for NGS according to the procedures described above.

## Molecular signatures of coding and noncoding hits

Hits for targets with zero, one and two mismatches were merged and divided in two subsets according to the annotation, consisting of protein coding hits and noncoding hits, respectively. For the coding hits gene set, a set of 159 non-redundant genes was created from all hits with target sites in protein-coding genes. The frequency of each gene was determined. For the remaining hits, the five closest genes upstream and the five closest genes downstream of the target site were determined using GRCh.93 Ensembl gene data. The starting position of a gene and the starting position of a target site was taken as measure for proximity. A noncoding-hits gene set of 1,805 non-redundant genes was created and the frequency of each gene was determined. Overlaps between both gene sets and all gene sets in the Molecular Signatures Database (MSigDB) were computed using the MiSigDB Web Application to Investigate Gene Sets with a FDR q-value below 0.05

(*Subramanian et al., 2005*; *Liberzon et al., 2011*), Heatmaps with the top four overlapping gene sets were created using the resulting Excel tables (*Figure 5H*, *Supplementary files 13–14* and the Python visualization library seaborn 0.9.0 (*Waskom et al., 2014*).To collect interaction data, we searched the 25 genes that were associated with 'down in ERCC3 mutated cells' in the String 10.5 database by choosing up to five interactors in the first and second shell (*Szklarczyk et al., 2017*) (*Figure 5I*).

## Data availability

NGS data are provided as raw read count tables as *Supplementary files 1,* and *17–23*. Please note, raw read count tables associated with TGW and the three oTGW libraries are available from Dryad, https://doi.org/10.5061/dryad.rs432pr. Plasmids encoding oTGW 3Cs-gRNA libraries will be available through the Goethe University Depository (http://www.innovectis.de/INNOVECTIS-Frankfurt/Technologieangebote/Depository).

## Code availability

Custom software is publicly available from GitHuB, https://github.com/GEG-IBC2/3Cs (*GEG-IBC2, 2019*; copy archived at https://github.com/elifesciences-publications/3Cs).

## DNA oligonucleotides

DNA oligonucleotides were purchased from Sigma-Aldrich and Integrated DNA Technologies (IDT) as single or pooled oligonucleotides, and from Twist Bioscience or CustomArray Inc. as oligonucleotide pools. A detailed list of all oligonucleotides can be found as supplementary information (*Supplementary file 16*).

## Acknowledgements

We thank Ian Cheeseman and Kara Lavidge McKinley for providing Cas9-inducible hTERT–RPE1 cells. We thank Tony Gutschner and Christian Münch for critical reading and commenting on the manuscript. We thank Tobias Schmidt for valuable advice and support with NGS. The hybridoma (12G10, alpha-tubulin) developed by the University of Iowa was obtained from the Developmental Studies Hybridoma Bank, created by the NICHD of the NIH and maintained at The University of Iowa, Department of Biology, Iowa City, IA 52242, USA. This work was supported by the Hessian Ministry for Science and the Arts (HMWK, LOEWE-CGT, IIIL5-518/17.004), the German Research Foundation (DFG; CEF-MC - EXC115/2; ECCPS - EXC147/2) and in part by the LOEWE Center Frankfurt Cancer Institute (FCI) funded by the Hessen State Ministry for Higher Education, Research and the Arts (IIIL5-519/03/03.001-(0015)).

## Additional information

### Competing interests

Ivan Dikic: is co-founder, shareholder and CEO of Vivlion GmbH in Gründung. Also a senior editor of *eLife*. Martin Wegner: The Goethe University Frankfurt has filed a patent related to this work on which MW is an inventor (WO2017EP84625). Valentina Diehl: The Goethe University Frankfurt has filed a patent related to this work on which VD is an inventor (WO2017EP84625). Rahel de Bruyn: The Goethe University Frankfurt has filed a patent related to this work on which RDB is an inventor (WO2017EP84625). Svenja Wiechmann: The Goethe University Frankfurt has filed a patent related to this work on which SW is an inventor (WO2017EP84625). Andreas Ernst: The Goethe University Frankfurt has filed a patent related to this work on which AE is an inventor (WO2017EP84625). Manuel Kaulich: The Goethe University Frankfurt has filed a patent related to this work on which MK is an inventor (WO2017EP84625). Also co-founder, shareholder and CSO of Vivlion GmbH in Gründung. The other authors declare that no competing interests exist.

## Funding

| Funder | Grant reference number | Author |
|---|---|---|
| Hessisches Ministerium für Wissenschaft und Kunst | IIIL5-518/17.004 | Manuel Kaulich |
| Deutsche Forschungsge-meinschaft | EXC115/2 | Manuel Kaulich |
| Hessisches Ministerium für Wissenschaft und Kunst | IIIL5-519/03/03.001 | Manuel Kaulich |
| Deutsche Forschungsge-meinschaft | EXC147/2 | Manuel Kaulich |

The funders had no role in study design, data collection and interpretation, or the decision to submit the work for publication.

## Author contributions

Martin Wegner, Resources, Software, Formal analysis, Validation, Investigation, Visualization, Methodology, Writing—original draft, Writing—review and editing; Valentina Diehl, Formal analysis, Validation, Investigation, Visualization, Methodology, Writing—original draft; Verena Bittl, Investigation, Visualization, Writing—original draft; Rahel de Bruyn, Investigation, Visualization, Methodology; Svenja Wiechmann, Investigation, Methodology; Yves Matthess, Investigation, Methodology, Writing—original draft, Writing—review and editing; Marie Hebel, Software, Investigation, Visualization, Methodology; Michael GB Hayes, Sven Heinz, Resources, Investigation; Simone Schaubeck, Validation, Investigation; Christopher Benner, Resources, Software; Anja Bremm, Investigation, Writing—original draft; Ivan Dikic, Supervision, Funding acquisition, Investigation, Writing—original draft, Writing—review and editing; Andreas Ernst, Conceptualization, Methodology, Writing—original draft, Writing—review and editing; Manuel Kaulich, Conceptualization, Software, Formal analysis, Supervision, Funding acquisition, Validation, Investigation, Visualization, Methodology, Writing—original draft, Project administration

## Author ORCIDs

Martin Wegner http://orcid.org/0000-0001-6403-3926
Yves Matthess http://orcid.org/0000-0003-4040-1258
Anja Bremm http://orcid.org/0000-0003-1386-0926
Ivan Dikic https://orcid.org/0000-0001-8156-9511
Manuel Kaulich http://orcid.org/0000-0002-9528-8822

## Decision letter and Author response

Decision letter https://doi.org/10.7554/eLife.42549.043
Author response https://doi.org/10.7554/eLife.42549.044

# Additional files

## Supplementary files

• Supplementary file 1. 3Cs-gRNA GFP library - NGS analysis including total read counts.
DOI: https://doi.org/10.7554/eLife.42549.016

• Supplementary file 2. List of 4–9N 3Cs libraries - NGS analysis including total read counts per library.
DOI: https://doi.org/10.7554/eLife.42549.017

• Supplementary file 3. 3Cs-gRNA DUB library - NGS analysis including total read counts of DUB library.
DOI: https://doi.org/10.7554/eLife.42549.018

• Supplementary file 4. 3Cs-gRNA DUB screen - NGS analysis including total read counts and normalizations.
DOI: https://doi.org/10.7554/eLife.42549.019

• Supplementary file 5. 3Cs-gRNA DUB screen - NGS analysis using MAGeCK.

DOI: https://doi.org/10.7554/eLife.42549.020

• Supplementary file 6. 3Cs-gRNA DUB screen - NGS analysis using PinAPL-Py.
DOI: https://doi.org/10.7554/eLife.42549.021

• Supplementary file 7. List of 3Cs-shRNA E2 library sequences and total NGS read counts.
DOI: https://doi.org/10.7554/eLife.42549.022

• Supplementary file 8. Number of TGW and oTGW target sequences per human chromosome and total oTGW NGS read counts per library.
DOI: https://doi.org/10.7554/eLife.42549.023

• Supplementary file 9. 3Cs TGW library – list of total NGS read counts.
DOI: https://doi.org/10.7554/eLife.42549.024

• Supplementary file 10. List of 4232 3Cs TGW gRNA sequences derived from the doxorubicin screen.
DOI: https://doi.org/10.7554/eLife.42549.025

• Supplementary file 11. 3Cs-gRNA TGW validation screen - NGS analysis including raw read counts, normalizations and ratios.
DOI: https://doi.org/10.7554/eLife.42549.026

• Supplementary file 12. Annotation list of validated coding TGW hits.
DOI: https://doi.org/10.7554/eLife.42549.027

• Supplementary file 13. Annotation list of validated noncoding TGW hits.
DOI: https://doi.org/10.7554/eLife.42549.028

• Supplementary file 14. List of coding hits - molecular signature analysis.
DOI: https://doi.org/10.7554/eLife.42549.029

• Supplementary file 15. List of noncoding hits - molecular signature analysis.
DOI: https://doi.org/10.7554/eLife.42549.030

• Supplementary file 16. List of DNA oligonucleotides.
DOI: https://doi.org/10.7554/eLife.42549.031

• Supplementary file 17. Raw sequencing counts of the six randomized libraries.
DOI: https://doi.org/10.7554/eLife.42549.032

• Supplementary file 18. DUB library raw sequencing counts.
DOI: https://doi.org/10.7554/eLife.42549.033

• Supplementary file 19. DUB screen raw sequencing read counts, includes day 0, day 11, and day 21 for two replicates.
DOI: https://doi.org/10.7554/eLife.42549.034

• Supplementary file 20. Raw sequencing counts of the E2-shRNA library.
DOI: https://doi.org/10.7554/eLife.42549.035

• Supplementary file 21. Raw sequencing reads of the TGS screen triplicates.
DOI: https://doi.org/10.7554/eLife.42549.036

• Supplementary file 22. Raw sequencing reads of the TGW screen validation library.
DOI: https://doi.org/10.7554/eLife.42549.037

• Supplementary file 23. Raw sequencing reads of CTRL and treatment samples of the TGW validation screen.
DOI: https://doi.org/10.7554/eLife.42549.038

• Transparent reporting form
DOI: https://doi.org/10.7554/eLife.42549.039

### Data availability

All data generated or analysed during this study are included in the manuscript, supplementary files or are available through GitHub or Dryad. NGS data and custom software is available as supplementary files and from Dryad and GitHub. Plasmids encoding oTGW 3Cs-gRNA libraries will be made available through the Goethe University Depository (http://innovectis.de/technologien/goethe-depository/).

The following dataset was generated:

| Author(s) | Year | Dataset title | Dataset URL | Database and Identifier |
|---|---|---|---|---|
| Wegner M, Diehl V, Bittl V, de Bruyn R, Wiechmann S, Matthess Y | 2019 | Data from: Circular synthesized CRISPR/Cas gRNAs for functional interrogations in the coding and noncoding genome | https://dx.doi.org/10.5061/dryad.rs432pr | Dryad Digital Repository, 10.5061/dryad.rs432pr |

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
