## [Decision Letter]

Thank you for submitting your article "High-fidelity circular synthesized CRISPR/Cas gRNAs for functional interrogations in the coding and noncoding genome" for consideration by *eLife*. Your article has been reviewed by three peer reviewers, and the evaluation has been overseen by a guest Reviewing Editor and Jessica Tyler as the Senior Editor. The following individual involved in the review of your submission has agreed to reveal his identity: Michael R Schlabach (Reviewer #2).

The reviewers have discussed the reviews with one another and the Reviewing Editor has drafted this decision to help you prepare a revised submission. Given the nature of this contribution the editors recommend this be reconsidered as a Tools and Resources paper rather than as a Research Article.

Summary:

In this manuscript, Wegner et al. describe an innovative approach to generate pooled sgRNA or shRNA libraries. Such libraries are now ubiquitously used as part of functional genomics efforts and thus an improved method to generate them is of high significance and utility.

Essential revisions:

The paper has now been seen by three reviewers. There was broad enthusiasm about the approach for generating complex libraries. However, there were concerns about some aspects of the work and a general feeling that the paper would benefit by streamlining the presentation to focus on the most solid and novel aspects of the method.

Four areas of concern are highlighted below:

1) There was general enthusiasm for the library generation strategy and as well as the value of an alternate approaches to create high diversity complex libraries. That said there were concerns that the comparison to alternate PCR-based approaches may not fairly represent how well those approaches can work when carefully implemented. Although there were some differences on this point, it seems that the most conservative and time efficient way to respond to these concerns would be to generally emphasize the capabilities of the present strategy and avoid relative comparisons unless they are directly substantiated by side-by-side comparisons. Additionally, since this is primarily a methods paper, it was felt that a better description of the workflow, including time and reagent requirements would be important to address in revision. For example, in the text or as a table/figure, the workflow Figure 1A could be fleshed out to provide more specifics about time and yields.

2) There was concern that, while the underlying quality of the DUB screen may be high, that the analysis strategy is not well validated. The raw sequencing data output of the screen should be analyzed by one or more of the established hit-calling methods (e.g. MAGeCK) to provide a better sense of the robustness of the hits as well as the real world performance of this library.

3) There is considerable concern about the conclusions from the TGW screens. While there is an appreciation for the innovative nature of the approach, the quality of the results, both in terms of the completeness and false positive rates, is difficult to evaluate given the data presented. Short of what would likely be a major experimental and analytical effort, the authors should tone down the claims and present this as an exploratory effort. For example, limitations such as very low reproducibility and the need to allow multiple mismatches to map sgRNAs should be discussed alongside the need for extensive validation.

I also pass on a couple of the specific reviewers' comments to provide context for the above:

Regarding analysis of the DUB screen: The section on the DUB screen is a bit convoluted and would benefit from using standardized analysis methods and a more focused discussion of the screen results for example by omitting the GO term analysis, which does not add very much.

To analyze the data, the authors sum sgRNA counts overall genes, but the more established approach in the field would be to calculate phenotypes for each sgRNA and to then aggregate these into gene-level phenotypes and derive p-values using statistical tests. Standardized pipelines exist to perform these analyses, such as MAGeCK. These phenotypes should also be compared to those from negative controls, both non-targeting controls and neutral-cutting controls, in particular given that the cell line used by the authors is p53-positive.

In addition, the assessment of screen quality by read counts across replicates is not fair; this should be a comparison of sgRNA enrichments across replicates. The variation (and correlation) in read counts stems almost entirely from differences in the starting abundances of sgRNAs in the library. Finally, Figure 3—figure supplement 1G should perhaps just be a scatter plot of the two sets of phenotypes against each other; the current representation is biased by the sorting and makes the data look better than they might be.

Regarding the TGW screen: The tgw screens should be de-emphasized on results and a more frank description of the shortcomings of it as an exploratory technique, assuming they don't want to do more work to figure out what any of these guides are really doing. That was my major objection with the manuscript.

---

## [Author Response]

Four areas of concern are highlighted below:1) There was general enthusiasm for the library generation strategy and as well as the value of an alternate approaches to create high diversity complex libraries. That said there were concerns that the comparison to alternate PCR-based approaches may not fairly represent how well those approaches can work when carefully implemented. Although there were some differences on this point, it seems that the most conservative and time efficient way to respond to these concerns would be to generally emphasize the capabilities of the present strategy and avoid relative comparisons unless they are directly substantiated by side-by-side comparisons. Additionally, since this is primarily a methods paper, it was felt that a better description of the workflow, including time and reagent requirements would be important to address in revision. For example, in the text or as a table/figure, the workflow Figure 1A could be fleshed out to provide more specifics about time and yields.

Area#1 (relative comparisons):We thank the reviewers for pointing out that we have made relative comparisons. In the revised version of the manuscript, we avoid comparable statements where not appropriate. As a result, the text has been reworded and Supplementary Figure 3C showing NGS comparisons of previously reported reagents with our 3Cs-DUB library has been removed.

Area#2 (visualization of workflow):The reviewers pointed out that a better visualization of the workflow that includes times, yields and reagent requirements is necessary. As this is primarily a methods paper, we are thankful for this critical comment and provide a new Figure 1A in which the molecular biology of 3Cs reactions is put into context of phage, ssDNA and final library generation. We highlight possible break points in the protocol, at which users can pause or reagents be long-term stored. Furthermore, to better illustrate the general time frame, we provide time estimates for total and hands-on time and, when appropriate, provide reagent yields. In addition, we now provide a comprehensive list of all reagents and equipment required to perform 3Cs in the Materials and methods section of the revised manuscript, as well as in the key_resource_table.xlsx file.

2) There was concern that, while the underlying quality of the DUB screen may be high, that the analysis strategy is not well validated. The raw sequencing data output of the screen should be analyzed by one or more of the established hit-calling methods (e.g. MAGeCK) to provide a better sense of the robustness of the hits as well as the real world performance of this library.

Area#3 (MAGeCK analysis of DUB data):To provide a better sense of the robustness of the hits as well as the real-world performance of the DUB library, the reviewers pointed out to use the well-established MAGeCK algorithm. We are thankful for raising this critical point and followed the reviewer’s suggestion. As such, we have extensively adapted text and Figure 3 to reflect log2-fold changes and their associated p-values for individual proliferative DUB phenotypes. Even though not asked for, we provide shRNA-mediated validation of positive and negative proliferative DUB phenotypes as new experimental support for the performance of the gRNA library. Furthermore, one reviewer pointed out that our initial gene-ontology (GO) analysis does not add much to the conclusions drawn from this section. In light of the overall manuscript, we agree with the reviewer and have removed the GO analysis part from the text and Figure 3.

3) There is considerable concern about the conclusions from the TGW screens. While there is an appreciation for the innovative nature of the approach, the quality of the results, both in terms of the completeness and false positive rates, is difficult to evaluate given the data presented. Short of what would likely be a major experimental and analytical effort, the authors should tone down the claims and present this as an exploratory effort. For example, limitations such as very low reproducibility and the need to allow multiple mismatches to map sgRNAs should be discussed alongside the need for extensive validation.

Area#4 (deemphasizing TGW screens):In order to put conclusions drawn from our TGW screens into a better perspective, the reviewers asked to tone down general conclusions drawn from this section and emphasize limitations as very low reproducibility, the need to allow mismatches in order to identify target sequences, and the need for extensive validations. We thank the reviewers for stressing this topic and have substantially changed the Results and Discussion section to better reflect the limitations associated with the TGW screens. Furthermore, we now highlight the fact that the TGW screen has been an exploratory effort, provide a more careful examination on the shortcomings of library and screen and make clear that doxorubicin resistance-associated target regions will need extensive future validations.

To better reflect the overall criticism and enable an unbiased evaluation of the 3Cs technology in the future, we also provide a shortened manuscript title “Circular synthesized CRISPR/Cas gRNAs for functional interrogations in the coding and noncoding genome” in which we avoid “high-fidelity”.